# PHASE-AWARE MEMORY THOUGHT FOR 3D MEDICAL IMAGE REPORT GENERATION

## ABSTRACT

Multi-phase 3D contrast-enhanced imaging is indispensable for clinical diagnosis, yet current vision–language models (VLMs) inadequately capture temporal dynamics across imaging phases, thereby limiting their reliability in automated medical report generation. We propose the *Phase-aware Memory Thought* (PhoT) framework, a novel paradigm that integrates temporal progression patterns in multi-phase CT with structured clinical reasoning. PhoT incorporates: (i) phase-aware pretraining to learn temporally aligned visual representations; (ii) parameter-efficient fine-tuning to adapt these representations for report generation; and (iii) a structured inference mechanism ("Phase of Thought") that leverages diagnostic templates to enhance clinical fidelity. We curate and evaluate PhoT on a large-scale dataset comprising 12,230 multi-phase CT series from 61,332 patient cases. Experimental results demonstrate that PhoT consistently outperforms strong baselines in both retrieval and report generation, achieving superior accuracy and interpretability. This work establishes PhoT as a clinically grounded, temporally aware VLM, advancing automated diagnostic reporting in complex medical imaging scenarios.

## 1 INTRODUCTION

3D contrast-enhanced imaging is integral to medical diagnostics, significantly enhancing anatomical and pathological visualization beyond what is achievable with non-contrast scans. This technique is particularly valuable in modalities such as computed tomography (CT) and magnetic resonance imaging (MRI), where the administration of contrast agents improves tissue differentiation and vascular visualization, thereby aiding in the detection and assessment of tumors Pandit et al. (2025), lesions Wei et al. (2024), and other abnormalities Liu et al. (2024). For a comprehensive clinical assessment, 3D contrast-enhanced imaging protocols often span multiple imaging planes (axial, sagittal, and coronal) and involve distinct acquisition phases: a pre-contrast scan, followed by contrast administration, and subsequent post-contrast scans timed to capture specific physiological processes Hsu et al. (2023). The interpretation of these complex, multi-phase datasets necessitates multidisciplinary collaboration Sack (2023), and while deep learning-based evaluation systems Huang et al. (2025a); Miller et al. (2024) are emerging to aid decision-making and streamline workflows, a unified approach for effectively leveraging the rich information across all imaging planes and temporal phases for diagnosis remains an ongoing challenge.

The automated generation of medical reports from such complex imaging data has become a pivotal research focus, aiming to reduce the substantial workload of radiologists and improve diagnostic consistency and efficiency. Initial efforts in this domain adapted image captioning techniques, employing encoder-decoder frameworks with Convolutional Neural Networks (CNNs), Recurrent Neural Networks (RNNs), or Transformers, often enhanced by attention mechanisms. The advent of Vision Transformers (ViTs) further advanced the field by enabling the capture of global contextual information. Concurrently, multi-modal approaches integrating diverse patient data and contrastive learning for improved visual-textual alignment have shown promise. Diagnostic reports themselves integrate spatial, temporal, and pathological correlations observed across multi-phase medical images, encoding rich clinical insights within textual narratives. Vision-language models (VLMs), such as BiomedCLIP Zhang et al. (2023) and LLaVA-Med Li et al. (2024a), have demonstrated strong capabilities in learning joint representations of visual and textual data. Building on this, early efforts like UniMedI He et al. (2024) leveraged reports as a shared semantic space for multi-modal medical images, while more recent approaches like fVLM Shui et al. (2025) directly associate 3D image-text

pairs. Despite these advancements, many current VLM-based methods focus mainly on spatial features and often overlook the critical temporal information inherent in multi-phase radiology, where dynamics such as contrast agent progression are crucial for clinical decision-making and are often detailed in reports. Aligning these reports with specific imaging phases offers a promising avenue to improve feature representations in 3D contrast-enhanced imaging (Figure 1).

In the broader computer vision (CV) domain, multi-view image analysis primarily addresses spatial and geometric relationships between images from different perspectives Wang et al. (2015), with applications in video understanding Siddiqui et al. (2024), 3D rendering Huang et al. (2025b), and segmentation Qin et al. (2023); Chen et al. (2025). Strategies like multi-scale feature aggregation Yu et al. (2024); Wang et al. (2023); Lin et al. (2023a); Cai et al. (2023) and sequential architectures (LSTMs Hong et al. (2023); Tang et al. (2024), Transformers Dong et al. (2023); Yang et al. (2023); Peng et al. (2022)) capture sequential or temporal dependencies Alkin et al. (2024); Chang et al. (2024). However, adapting these general CV techniques to the specific nuances of medical imaging—especially for aligning spatial, phase-specific temporal information, and textual clinical insights from 3D contrast-enhanced sequences—remains a challenging open problem. Beyond report generation, robust medical reasoning is paramount. This involves interpret-

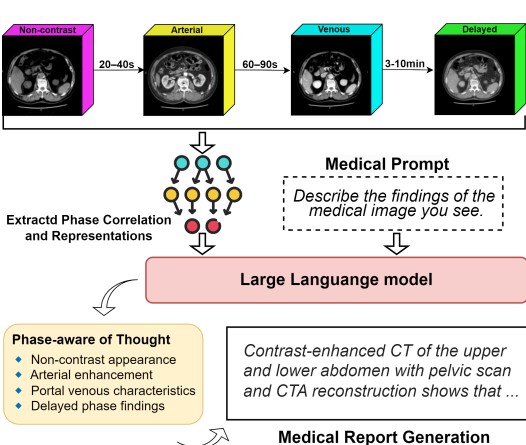

Figure 1: **Motivation.** Conventional report generation neglects temporal reasoning across medical phases. PhoT introduces structured reasoning to model phase correlations, improving the accuracy and clinical relevance of multiphase reports.

ing visual data to understand patient conditions, formulate diagnoses, and guide treatment, often requiring structured inference, the application of diagnostic criteria, and the integration of medical knowledge Rao et al. (2025); Kim et al. (2025). Current research explores simulating clinical reasoning Jiang et al. (2025) and incorporating external knowledge like graphs Liu et al. (2021); Wu et al. (2025), with a growing emphasis on Explainable AI (XAI) to ensure transparency and clinical trust.

Despite these significant advancements in medical report generation and reasoning, a critical gap exists in effectively integrating the temporal dynamics inherent in multi-phase 3D medical imaging with structured, clinically-relevant reasoning processes. Current models often struggle to explicitly capture and leverage the evolution of findings across different contrast phases, which is essential for accurate diagnosis and comprehensive reporting in many clinical scenarios. This limitation hinders the ability to generate reports that fully reflect the nuanced understanding a radiologist develops by observing these temporal changes. To address these limitations, we introduce the Phase-aware Memory Thought (PhoT) method, a novel framework designed to enhance medical image analysis by explicitly integrating temporal dynamics from multi-phase imaging sequences and employing a structured reasoning paradigm. Our contributions are threefold:

- **Phase-aware Medical Alignment.** We propose a new framework that learns unified representations across multi-phase medical images and diagnostic text by capturing temporal progression patterns in imaging sequences.
- **Phase-aware Diagnosis Generation.** We design an efficient fine-tuning strategy that preserves phase-level temporal understanding while enabling accurate and scalable report generation from complex 3D scans.
- **Phase of Thought Inference.** We introduce a structured inference paradigm that guides the model to reason over phase-series data using diagnostic templates, improving interpretability and clinical relevance of the generated reports.

The PhoT method's phase-aware pretraining captures robust temporal features, its efficient fine-tuning adapts these features for coherent report generation, and its structured inference mechanism ensures that the generated reports are not only accurate but also clinically insightful by systematically addressing diagnostic criteria based on the full multi-phase sequence.

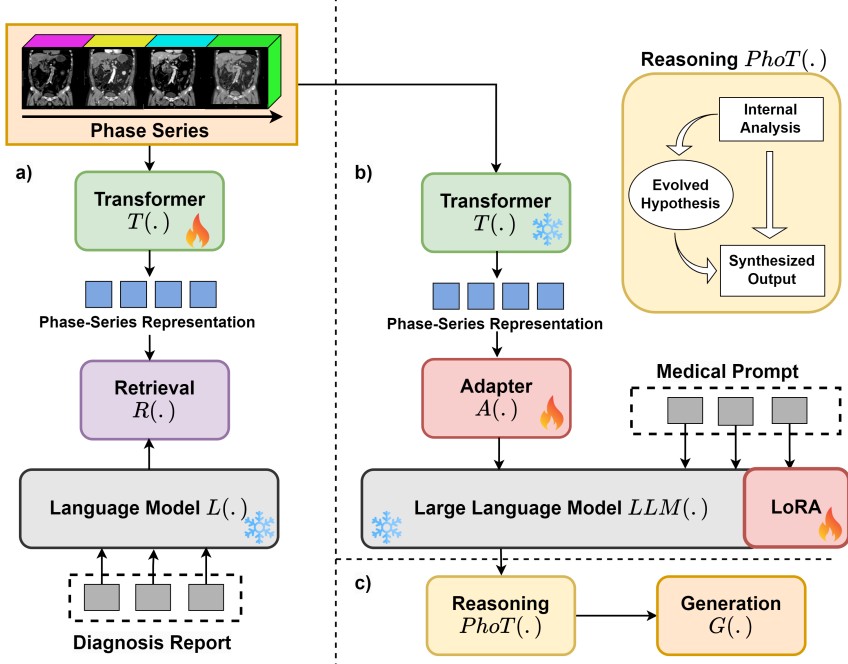

Figure 2: **Phase-aware memory thought (PhoT) framework.** (a) Phase-aware Pretraining integrates multi-phase imaging sequences into robust, temporal-aware visual representations through Vision Transformer-based multi-scale feature extraction and a gated memory update mechanism. (b) Phase-aware Fine-tuning leverages pretrained frozen visual encoders to adapt spatial features via a spatial adaptor, enabling efficient transfer for report generation tasks by aligning visual tokens with textual embeddings. (c) Phase of Thought (PhoT) employs structured diagnostic queries (caption templates) to systematically guide inference, synthesizing detailed, coherent medical reports from integrated multi-phase visual information.

## 2 RELATED WORK

### 2.1 MEDICAL REPORT GENERATION

Medical report generation transforms medical images (e.g., X-rays, MRIs, CT scans) into diagnostic textual reports, enhancing clinical decision-making by combining visual data interpretation with clinical notes and prior diagnostic information Zhang et al. (2024); Lu et al. (2024). Vision-language models (VLMs) are crucial for tasks such as image retrieval, classification, and explanatory reporting Reale-Nosei et al. (2024). For instance, BiomedCLIP learns joint representations from extensive biomedical image-text datasets, while LLaVA-Med fine-tunes models using biomedical figures and GPT-4-generated instructions to enhance semantic precision Zhang et al. (2023); Li et al. (2024a). However, generating reports from 3D medical imaging faces significant challenges due to limited annotated 3D data, arising from high expert labeling costs relative to 2D datasets Lin et al. (2023b). To mitigate this, methods like knowledge distillation (e.g., Self-evolving Vision Transformer Park et al. (2022)) and 2D slice extraction (e.g., UniMedI He et al. (2024)) have been developed. Recent models, such as fVLM, utilize proprietary 3D datasets but primarily focus on spatial features, overlooking temporal dynamics like contrast progression Shui et al. (2025). Addressing this gap, we introduce a curated multi-phase CT dataset to enable temporally-aware report generation, thus improving diagnostic accuracy.

### 2.2 LANGUAGE MODEL REASONING

Language model reasoning has become increasingly significant, demonstrating substantial potential across numerous domains. Specifically, the Chain-of-Medical-Thought (CoMT) approach highlights Chain-of-Thought (CoT) reasoning in medical image analysis and report generation, effectively

reducing hallucinations and simulating expert diagnostic processes Jiang et al. (2025). Despite advancements, exploiting language models' reasoning capacities in medicine remains underexplored, particularly in complex scenarios like multi-phase imaging Rao et al. (2025); Kim et al. (2025). Research into broader reasoning strategies, such as the Deductive and InDuctive (DID) methodology Cai et al. (2024b), advocates integrating structured reasoning processes Wei et al. (2022); Yao et al. (2023). These cognitive-inspired approaches emphasize dynamic combinations of deductive and inductive reasoning, enhancing model adaptability and efficacy in complex problem-solving contexts Cai et al. (2024a); Besta et al. (2024). Collectively, these studies indicate that refining reasoning frameworks within language models is essential for advancing their effectiveness and reliability in intricate medical applications.

## 3 METHOD

In this section, we present the Phase-aware Memory Thought (PhoT), which which enhances medical image analysis by integrating temporal dynamics across multi-phase imaging sequences (Figure 2).

### 3.1 PHASE-AWARE PRETRAINING

PhoT pretrains a visual encoder to predict robust feature representations for cross-modal retrieval, leveraging multi-phase medical imaging sequences by integrating temporal dynamics across phases.

**Input and Multi-scale Feature Extraction.** Each phase $t$ in a multi-phase sequence is represented as a 3D tensor $I_t \in \mathbb{R}^{C \times D \times H \times W}$, where $C$ denotes the channel count and $D \times H \times W$ the spatial dimensions. The input is segmented into non-overlapping patches and processed by a Vision Transformer (ViT) to derive feature embeddings:

$$F_t = \Psi(I_t), \quad F_t \in \mathbb{R}^{N \times C'} \tag{1}$$

Here, $\Psi(\cdot)$ encapsulates ViT's patch embedding and contextualization, with $N$ patches and embedding dimension $C'$. To capture varied spatial contexts, convolutions with kernel sizes $K = \{k_1, k_2, \ldots, k_m\}$ are applied to $F_t$, reshaping features into spatial maps, convolving, and reshaping back:

$$\hat{F}_{t,k_i} = \Phi_{F,k_i}(F_t), \quad M_{t-1,k_i} = \Phi_{M,k_i}(M_{t-1}) \tag{2}$$

where $\hat{F}_{t,k_i}, M_{t-1,k_i} \in \mathbb{R}^{N \times C'}$, enhancing feature robustness via multi-scale representations.

**Phase Aggregation with Memory Update.** A gating mechanism aggregates temporal information by maintaining a memory state $M_t$, updated dynamically with phase features $F_t$. Learnable update ($z_t$) and reset ($r_t$) gates regulate the balance between new and prior information:

$$z_t = \sigma(\Phi_F^{(z)}(F_t) + \Phi_M^{(z)}(M_{t-1})) \odot \theta_z, \quad r_t = \sigma(\Phi_F^{(r)}(F_t) + \Phi_M^{(r)}(M_{t-1})) \odot \theta_r \tag{3}$$

Here, $\sigma(\cdot)$ is the sigmoid function, $\Phi_F^{(z)}, \Phi_M^{(z)}, \Phi_F^{(r)}, \Phi_M^{(r)}$ are learnable convolutional transformations, and $\theta_z, \theta_r \in \mathbb{R}^N$ are scaling parameters. A candidate memory state $M_c$ integrates multi-scale features and prior memory, modulated by the reset gate:

$$M_c = \sum_{i=1}^{m} \tanh(\hat{F}_{t,k_i} + r_t \odot M_{t-1,k_i}) \tag{4}$$

The memory state is updated by blending prior and candidate states under the update gate's guidance:

$$M_t = (1 - z_t) \odot M_{t-1} + z_t \odot M_c \tag{5}$$

At the final phase $T$, the memory state $M_T \in \mathbb{R}^{N \times C'}$ is aggregated into a compact representation using attention pooling, which weights patches by their importance and combines them into a single

vector $\bar{M}_T \in \mathbb{R}^{C'}$. This vector is projected into a shared vision-language space and L2-normalized as $\hat{M}$.

**Text Encoding and Contrastive Alignment.** Textual descriptions are processed by a pretrained language model (e.g., BERT) to produce contextualized embeddings $\hat{T} \in \mathbb{R}^{C'}$, which are L2-normalized. To align image embeddings $\hat{M}$ with text embeddings $\hat{T}$, a contrastive loss is employed. For a batch of $B$ image-text pairs, the similarity matrix is computed as $S = \mathbf{M}^\top \cdot \mathbf{T} \in \mathbb{R}^{B \times B}$, where $\mathbf{M}, \mathbf{T} \in \mathbb{R}^{C' \times B}$ contain L2-normalized image and text embeddings, respectively. The contrastive loss is defined as:

$$\mathcal{L} = -\frac{1}{B} \sum_{i=1}^{B} \log \frac{\exp(S_{i,i}/\tau)}{\sum_{j=1}^{B} \exp(S_{i,j}/\tau)} \tag{6}$$

where $\tau$ is a learnable temperature parameter. This loss maximizes similarity for matching image-text pairs while minimizing it for non-matching pairs.

## 3.2 PHASE-AWARE FINE-TUNING

Building upon the robust feature representations learned during pretraining, the fine-tuning stage adapts the model for generating textual reports from multi-phase 3D medical imaging sequences. Notably, the parameters of the pretrained visual encoder are frozen, preserving its temporal aggregation capabilities while focusing optimization on the spatial adaptor and language model.

**Frozen Visual Encoder.** The visual encoder, trained to capture temporal dynamics across imaging phases, serves as a fixed feature extractor during fine-tuning. Its parameters remain unchanged, ensuring that the memory state $M_T \in \mathbb{R}^{N \times C'}$ retains the multi-scale, phase-aware representations learned in pretraining. This approach reduces computational overhead and leverages the encoder's established ability to handle complex imaging sequences.

**Spatial Adaptor Mechanism.** Given the frozen visual encoder, the spatial adaptor transforms the memory state $M_T$ into a form suitable for the language model. A downsampling operator $\mathcal{D}_k$, parameterized by a factor $k$, aggregates features over local spatial regions:

$$M_T^{\text{pooled}} = \mathcal{D}_k(M_T) \in \mathbb{R}^{N' \times C'}, \tag{7}$$

where $N' < N$ reflects a reduced token count. This is achieved by averaging features within spatially adjacent groups of $k \times k \times k$ patches in the 3D grid, preserving the channel dimension $C'$. A learnable linear projection $\phi : \mathbb{R}^{C'} \to \mathbb{R}^{C_{\text{LLM}}}$ then aligns these features with the language model's embedding space:

$$\hat{M}_T = \phi(M_T^{\text{pooled}}) \in \mathbb{R}^{N' \times C_{\text{LLM}}}. \tag{8}$$

This mechanism ensures computational efficiency while maintaining the spatial and feature information necessary for report generation.

**Language Model Fine-tuning for Report Generation.** The adapted visual tokens $\hat{M}_T$ are integrated into the language model's input sequence, which generates the report $R = (r_1, r_2, \ldots, r_L)$ autoregressively. The model is optimized using the cross-entropy loss:

$$\mathcal{L}_{\text{LM}} = -\sum_{i=1}^{L} \log P(r_i | \hat{M}_T, r_{<i}), \tag{9}$$

where $P(r_i | \hat{M}_T, r_{<i})$ is the probability of the $i$-th token given the visual tokens and preceding tokens $r_{<i}$. Only the spatial adaptor parameters are updated during training, allowing the model to specialize for report generation while leveraging the frozen visual encoder's representations and the language model's capability.

### 3.3 PHASE OF THOUGHT (PHOT)

PhoT performs structured inference for medical report generation, leveraging a predefined set of caption templates $\mathcal{Q} = \{q_1, q_2, \ldots, q_K\}$. Each template $q_k$ directs the model to systematically examine phase-series data according to specific diagnostic criteria, facilitating detailed analysis and coherent synthesis.

**Internal Analysis.** For each template $q_k$, the model computes latent observations $\mathcal{O}_k$ by integrating phase-specific features $\{F_t\}_{t=1}^T$ and the memory state $M_T$:

$$\mathcal{O}_k = \mathcal{A}_k(\{F_t\}_{t=1}^T, M_T, q_k), \tag{10}$$

where $\mathcal{A}_k(\cdot)$ represents the analysis function tailored to $q_k$. This step implicitly considers phase-specific imaging signatures across the temporal sequence—such as the baseline tissue appearance in non-contrast scans ($F_1$), vascular enhancement in arterial phases ($F_2$), organ parenchyma characteristics in portal venous phase ($F_3$)s, and delayed retention patterns ($F_4$), with temporal relationships encoded in $M_T$.

**Final Output Generation.** The latent observations $\{\mathcal{O}_k\}_{k=1}^K$ are synthesized into a single narrative output $R$, generated as:

$$R = \mathcal{S}(\{\mathcal{O}_k\}_{k=1}^K; \hat{M}_T), \tag{11}$$

where $\mathcal{S}(\cdot)$ is a synthesis function, and $\hat{M}_T$ is an adapted memory state derived from $M_T$. The output $R$ is a sequence of report tokens forming a cohesive paragraph that directly addresses the diagnostic question posed by $q_k$, without enumerating phase-specific details.

This formulation ensures that inference leverages the full phase-series $\{I_t\}_{t=1}^T$, with the caption templates driving a structured analysis and synthesis process.

## 4 EXPERIMENTS

**CT Phase Datasets.** We collected 61,332 CT cases in 2024 at Dongfang Hospital using the DISCOVERY CT750 HD FREEDOM system under standardized protocols. Anatomical regions included head, chest, abdomen, pelvis, spine, soft tissues, vasculature, and joints. After grouping into pre-contrast and contrast-enhanced phase series, the dataset comprised 12,230 samples (7,142 two-phase, 3,451 three-phase, and 1,637 four-phase).

**Implementation Details.** Experiments were run on an Inspur NF5468M6 server with 8×A100 GPUs using DeepSpeed bf16 training. Images were normalized and resized to $32 \times 256 \times 256$; text inputs were capped at 512 tokens. A 12-layer 3D ViT encoder fed into a pretrained BERT, with only a lightweight 3D adapter fine-tuned. Optimization used AdamW with warm-up and cosine decay. Retrieval was measured with Recall@k, and report generation with BLEU, ROUGE-1, METEOR, and BERT-F1. Further implementation details are provided in the Appendix.

### 4.1 RESULTS

#### 4.1.1 MEDICAL RETRIEVAL

Table 1 compares a Vanilla Baseline, 2D models (PMC-CLIP Lin et al. (2023b), MedCLIP Wang et al. (2022), BiomedCLIP Zhang et al. (2023)), 3D models (BUID Cao et al. (2024), ASG Li et al. (2024b), CT-GLIP Lin et al. (2024)), and the proposed PhoT.

**Baseline 3D CLIP Shows Competitive Performance.** The Vanilla Baseline, a 3D CLIP model on which PhoT is developed, performs competitively, rivaling 3D models. At K = 100, it achieves IR R@1 of 18.00 and TR R@10 of 63.00, but recall drops at K = 2000 (IR R@1 = 3.10, TR R@10 = 12.65). Comparable to BUID (IR R@10 = 11.55 at K = 2000) and ASG (TR R@10 = 14.55), its contrastive learning captures 3D features effectively, though it lags behind PhoT in complex tasks.

Table 1: Performance comparison of retrieval models across retrieval set sizes (K). IR R@k and TR R@k denote Image and Text Retrieval Recall at k. Best results are in **bold**.

| K | Metric | Vanilla | 2D Models | | | 3D Models | | | Proposed |
|---|---|---|---|---|---|---|---|---|---|
| | | Baseline | PMC-CLIP | MedCLIP | BiomedCLIP | BUID | ASG | CT-GLIP | PhoT |
| 100 | IR R@1 | 18.00 | 2.00 | 2.00 | 1.00 | 17.00 | 19.00 | 16.00 | **21.00** |
| | IR R@5 | 42.00 | 9.00 | 9.00 | 7.00 | 46.00 | 41.00 | 46.00 | **56.00** |
| | IR R@10 | 65.00 | 14.00 | 17.00 | 15.00 | 67.00 | 64.00 | 65.00 | **73.00** |
| | TR R@1 | 15.00 | 1.00 | 3.00 | 2.00 | **16.00** | **16.00** | 14.00 | 12.00 |
| | TR R@5 | 46.00 | 7.00 | 10.00 | 10.00 | 40.00 | 48.00 | 47.00 | **53.00** |
| | TR R@10 | 63.00 | 13.00 | 24.00 | 19.00 | 63.00 | 65.00 | 65.00 | **75.00** |
| 500 | IR R@1 | 6.30 | 0.60 | 0.20 | 1.00 | 6.50 | 6.50 | 6.40 | **9.00** |
| | IR R@5 | 20.20 | 2.20 | 2.40 | 3.40 | 20.30 | 19.70 | 19.60 | **25.60** |
| | IR R@10 | 31.30 | 4.60 | 4.00 | 5.00 | 31.60 | 30.90 | 33.10 | **36.60** |
| | TR R@1 | 8.00 | 0.40 | 0.60 | 0.60 | 6.00 | 5.70 | 6.30 | **8.60** |
| | TR R@5 | 20.60 | 1.60 | 2.80 | 2.60 | 20.80 | **22.00** | 21.40 | **25.00** |
| | TR R@10 | 29.80 | 3.60 | 5.40 | 5.20 | 30.00 | 29.60 | 30.50 | **36.60** |
| 1000 | IR R@1 | 3.60 | 0.10 | 0.10 | 0.40 | 3.90 | 3.50 | 4.00 | **5.50** |
| | IR R@5 | 11.60 | 1.40 | 1.00 | 1.40 | 11.40 | **12.20** | 12.10 | **18.10** |
| | IR R@10 | 20.80 | 1.90 | 1.80 | 2.70 | 19.20 | 20.40 | 22.00 | **26.50** |
| | TR R@1 | 4.20 | 0.10 | 0.30 | 0.30 | 3.40 | 3.30 | 4.10 | **6.20** |
| | TR R@5 | 12.20 | 0.90 | 1.10 | 1.10 | 12.30 | **12.50** | 12.30 | **17.60** |
| | TR R@10 | 19.20 | 1.80 | 3.00 | 2.30 | 18.90 | **19.60** | 19.00 | **26.00** |
| 2000 | IR R@1 | 3.10 | 0.15 | 0.05 | 0.20 | 2.30 | 2.35 | 2.40 | **4.10** |
| | IR R@5 | 8.40 | 0.50 | 0.50 | 0.40 | 8.44 | **8.45** | 8.35 | **12.35** |
| | IR R@10 | 12.85 | 0.85 | 0.90 | 1.30 | 11.55 | 12.70 | **13.80** | **17.70** |
| | TR R@1 | 2.55 | 0.15 | 0.20 | 0.25 | **2.70** | 2.30 | 2.60 | **4.00** |
| | TR R@5 | 7.35 | 0.35 | 1.00 | 0.65 | **8.30** | 7.60 | 7.10 | **11.90** |
| | TR R@10 | 12.65 | 0.80 | 1.55 | 1.30 | 12.25 | **14.55** | 13.95 | **18.15** |

**2D Models Lack Spatial Awareness.** 2D models (PMC-CLIP, MedCLIP, BiomedCLIP) under-perform significantly. At K = 100, PMC-CLIP's IR R@1 is 2.00, dropping to 0.15 at K = 2000; MedCLIP and BiomedCLIP fare similarly (IR R@1 = 0.05, 0.20). Their inability to model 3D spatial relationships results in poor embeddings, underscoring the need for 3D architectures in volumetric imaging tasks.

**3D Models Outperform 2D Counterparts.** 3D models (BUID, ASG, CT-GLIP) outperform 2D counterparts by leveraging spatial awareness. At K = 100, ASG's IR R@10 is 64.00 and BUID's IR R@10 is 67.00, compared to BiomedCLIP's 15.00. At K = 2000, CT-GLIP's IR R@10 (13.80) exceeds PMC-CLIP's 0.85. However, performance plateaus (e.g., IR R@1 = 2.30–4.00), trailing PhoT.

Table 2: Report Generation Results.

| Test | Metric | Model | | | | | |
|---|---|---|---|---|---|---|---|
| | | Merlin | fVLM | Baseline | PhoT-NoT | CoT | PhoT |
| 100 | BLEU | 12.19 | 12.85 | 12.44 | 12.93 | 12.85 | 13.86 |
| | ROUGE1 | 14.97 | 16.54 | 15.47 | 16.54 | 15.47 | 18.11 |
| | METEOUR | 11.67 | 12.47 | 13.33 | 13.86 | 13.33 | 14.27 |
| | BERTF1 | 72.56 | 75.68 | 75.54 | 82.68 | 73.04 | 83.81 |
| 500 | BLEU | 12.13 | 13.58 | 12.01 | 13.48 | 12.26 | 13.58 |
| | ROUGE1 | 15.38 | 17.07 | 15.57 | 17.07 | 15.81 | 17.27 |
| | METEOUR | 12.12 | 13.16 | 13.27 | 14.16 | 13.27 | 14.32 |
| | BERTF1 | 74.26 | 72.59 | 74.21 | 76.62 | 72.29 | 83.94 |
| 1000 | BLEU | 11.95 | 13.50 | 12.29 | 13.50 | 12.34 | 14.39 |
| | ROUGE1 | 15.20 | 17.14 | 16.01 | 17.14 | 16.01 | 18.05 |
| | METEOUR | 11.86 | 12.93 | 13.07 | 13.93 | 13.65 | 15.01 |
| | BERTF1 | 74.58 | 72.68 | 72.22 | 82.68 | 72.31 | 83.80 |
| 2000 | BLEU | 12.59 | 13.44 | 11.86 | 13.44 | 12.43 | 14.11 |
| | ROUGE1 | 15.87 | 17.03 | 15.71 | 17.03 | 16.11 | 17.98 |
| | METEOUR | 12.33 | 13.00 | 13.22 | 14.00 | 13.65 | 14.65 |
| | BERTF1 | 75.61 | 72.60 | 73.28 | 82.60 | 73.32 | 83.95 |

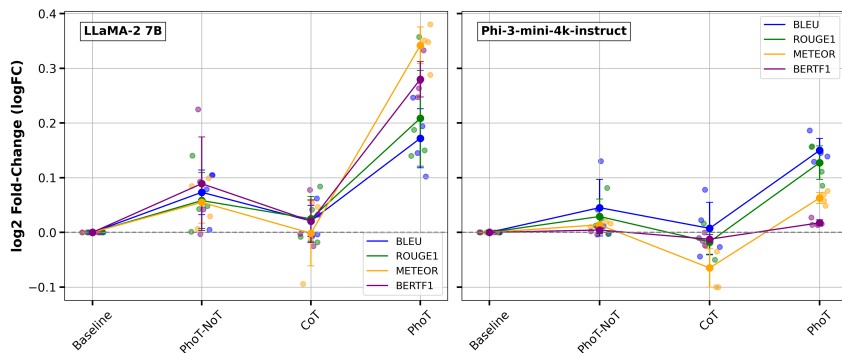

Figure 3: Fold-change over the Vanilla Baseline. **Left:** LLaMA-2. **Right:** Phi-3-mini-4k-instruct. Each point represents the logFC for a specific test size. Error bars denote the mean ± standard deviation of logFC across test sizes for each method and metric. A horizontal dashed line at logFC = 0 indicates no change relative to the baseline. Solid lines connect the mean logFC values for each metric across methods.

**PhoT Outperforms All Baselines.**    PhoT outperforms all models across most metrics. At K = 100, it achieves IR R@10 of 73.00 and TR R@10 of 75.00, surpassing the Baseline (65.00, 63.00). At K = 2000, PhoT's IR R@10 is 17.70 and TR R@10 is 18.15, exceeding CT-GLIP's 13.80 and 13.95. Its advanced feature alignment enhances image-text correspondence, making it ideal for clinical 3D retrieval.

### 4.1.2    REPORT GENERATION

Table 2 compares PhoT with established 3D models (Merlin Blankemeier et al. (2024), fVLM Shui et al. (2025)), a Vanilla Baseline (CLIP-based, lacking phase series support), and ablation variants (PhoT-NoT, CoT) for CT report generation.

**Baseline Performance Comparable to Existing Methods.**    The Vanilla Baseline demonstrates competitive performance, with scores comparable to those of Merlin and fVLM. For example, at test size 100, its BLEU score is 12.44, while Merlin's is 12.19 and fVLM's is 12.85. For BERTF1, the Baseline achieves 75.54, compared to Merlin's 72.56 and fVLM's 75.68, showing that it performs similarly to these established models. This suggests that the baseline's CLIP-based architecture effectively captures essential features for report generation, despite lacking phase series support, making it a robust foundation for further enhancements.

**PhoT Consistently Outperforms All Models.**    PhoT consistently surpasses Merlin, fVLM, and the Vanilla Baseline across all metrics and test sizes. At test size 100, PhoT achieves a BLEU score of 13.86, ROUGE1 of 18.11, METEOUR of 14.27, and BERTF1 of 83.81, significantly outperforming fVLM's 12.85, 16.54, 12.47, and 75.68, respectively. At test size 2000, PhoT's BLEU (14.11), ROUGE1 (17.98), METEOUR (14.65), and BERTF1 (83.95) remain superior, with BERTF1 notably higher than fVLM's 72.60. The incorporation of phase series enables PhoT to better model temporal and contextual relationships in CT data, resulting in enhanced report quality and semantic coherence, positioning it as a superior choice for clinical report generation.

**Impact of Phase Modeling and Inference.**    PhoT-NoT, a variant of PhoT without the thinking process, improves performance over the Baseline by leveraging phase series, e.g., achieving a BERTF1 of 82.68 at test size 100 versus the Baseline's 75.54. Conversely, CoT, another PhoT variant using a naive Chain of Thought approach, degrades performance, with a BERTF1 of 73.04 at test size 100, falling below the Baseline. However, PhoT, with its optimized inference on phase series, boosts performance significantly, reaching a BERTF1 of 83.81, highlighting the efficacy of a tailored thinking chain.

## 4.2 ABLATION STUDY

**Foundation Model.** We replaced the primary model with LLaMA-2 and Phi-3-mini-4k-instruct to test generalizability. As shown in Figure 3, PhoT consistently improved performance, PhoT-NoT achieved moderate gains, while the naive CoT variant showed little or negative impact, underscoring the value of PhoT's structured inference.

**Clinical Metrics.** In addition to standard NLG metrics (BLEU, ROUGE, METEOR, BERTF1), we incorporated clinically oriented metrics such as **GREEN** (Generative Radiology Report Evaluation and Error Notation) and a **Qwen-based LLM evaluation** to better assess clinical faithfulness. Table 3 presents results with 95% confidence intervals.

| Metric | Baseline | Merlin | fVLM | PhoT (Ours) |
|---|---|---|---|---|
| GREEN | $19.00 \pm 0.25$ | $19.50 \pm 0.25$ | $20.50 \pm 0.25$ | $\mathbf{21.60 \pm 0.22}$ |
| Qwen-based | $3.80 \pm 0.13$ | $3.90 \pm 0.13$ | $4.10 \pm 0.13$ | $\mathbf{4.31 \pm 0.13}$ |

Table 3: Clinical metric evaluation across models.

**Evaluation on CT-RATE.** To test generalization, we evaluated PhoT under zero-shot conditions on the **CT-RATE** dataset, which consists solely of non-contrast phase data. Table 4 shows performance comparisons against prior baselines, demonstrating PhoT's robust generalizability.

| Metric | CT-VocabFine | CT-LiPro | CT-CLIP | BIUD | Merlin | fVLM | PhoT |
|---|---|---|---|---|---|---|---|
| AUC | 75.0 | 75.1 | 70.4 | 71.3 | 72.8 | **77.8** | 77.5 |
| ACC | 60.2 | 67.6 | 65.1 | 68.1 | 67.2 | 71.8 | **72.6** |
| F1 | 72.8 | 71.4 | 69.1 | 71.6 | 70.9 | 75.1 | **76.2** |
| Prec | 34.2 | 33.1 | 30.6 | 33.8 | 33.7 | **37.9** | 36.4 |
| Spec | – | – | – | 68.6 | 66.8 | 71.7 | **72.2** |
| Sens | – | – | – | 67.3 | 70.1 | **72.8** | 71.9 |

Table 4: Evaluation of PhoT and baselines on CT-RATE dataset.

**Confidence Interval Analysis.** We computed mean, standard deviation (SD), and 95% confidence intervals (CIs) across multiple seeds and test set sizes (100, 500, 1000, 2000). This ensures statistical robustness and reproducibility. Table 5 summarizes the results for key metrics.

| Model | Metric | Mean | SD | 95% CI |
|---|---|---|---|---|
| Baseline | BLEU | 12.15 | 0.25 | [11.75, 12.55] |
| | ROUGE1 | 15.69 | 0.23 | [15.32, 16.06] |
| | METEOR | 13.22 | 0.11 | [13.04, 13.40] |
| | BERTF1 | 73.81 | 1.42 | [71.55, 76.07] |
| PhoT | BLEU | 13.99 | 0.35 | [13.43, 14.55] |
| | ROUGE1 | 17.85 | 0.37 | [17.26, 18.44] |
| | METEOR | 14.56 | 0.34 | [14.02, 15.10] |
| | BERTF1 | 83.88 | 0.07 | [83.77, 83.99] |

Table 5: Mean, SD, and 95% confidence intervals across seeds and test sets.

## 5 CONCLUSION

This study introduces PhoT, a phase-aware framework for 3D medical report generation that integrates temporal dynamics, efficient fine-tuning, and structured reasoning. PhoT improves retrieval, report accuracy, and interpretability. Broader validation would benefit from larger multi-phase datasets and standardized imaging protocols, but the framework nonetheless advances reasoning-enhanced diagnostic systems that emulate expert radiological workflows.

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

## A  APPENDIX

## B  DATASETS

### B.1  CT PHASE DATSETS

Computed Tomography (CT) images were acquired using the DISCOVERY CT750 HD FREEDOM system at Dongfang Hospital. In 2024, 61,332 patient cases were collected, adhering to standardized imaging protocols. Anatomical regions evaluated include head, chest, abdomen, pelvis, spine, soft tissues, vasculature, and joints. 3D images were grouped into phase-series: a pre-contrast phase and subsequent contrast-enhanced phases, yielding 12,230 series samples (7,142 two-phase, 3,451 three-phase, and 1,637 four-phase). 3D medical images are organized into a sequence of imaging phases reflecting contrast administration and clinical needs. Plane classification ensures a uniform axial, sagittal, or coronal view across each series, enabling precise temporal comparisons (Figure 4). The sequence starts with pre-contrast images capturing baseline anatomy without contrast, followed by contrast-enhanced phases—arterial, portal venous, and delayed—highlighting contrast dynamics to distinguish tissues and detect abnormalities like tumors or vascular issues. Next, dynamic phases such as Mean Transit Time (MTT), Cerebral Blood Volume (CBV), and Cerebral Blood Flow (CBF) offer functional insights into circulation and perfusion, distinct from structural contrast data. Post-contrast phases conclude the sequence, providing detailed diagnostic views. This structured approach aligns with clinical protocols, maintaining spatial and temporal coherence for accurate anatomical and functional assessment.

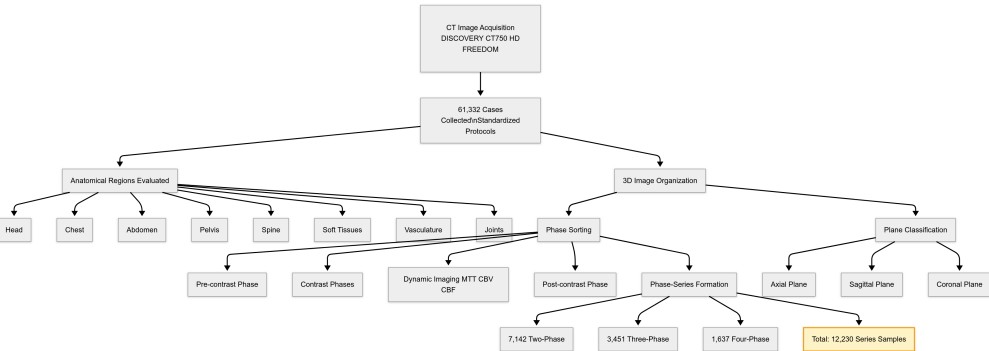

Figure 4: Flowchart of 3D CT Image Acquisition, Anatomical Evaluation, and Phase-Series Organization.

### B.2  DEMONSTRATION

**Axial Plane.**  We present an imaging phase series acquired in the axial plane, encompassing the non-contrast phase (Figure 5), arterial phase (Figure 6), venous phase (Figure 7), and delayed phase (Figure 8), accompanied by the following clinical caption:

*Findings: The liver is of normal size and contour. Two quasi-round hypodense lesions are identified in segments S4 and S5 of the liver, showing around 25 HU on non-contrast imaging, without significant enhancement following contrast administration. No dilatation of the bile ducts is observed. The gallbladder, pancreas, and spleen are unremarkable in morphology and density. Both kidneys exhibit multiple cystic lesions, which do not show contrast enhancement. No hydronephrosis or ureteral dilatation is noted. Adrenal glands appear unremarkable. No signs of obstruction or abnormal masses are detected in the abdominal intestines. There are patchy densities in the peritoneum, possibly indicating fat inflammation.*

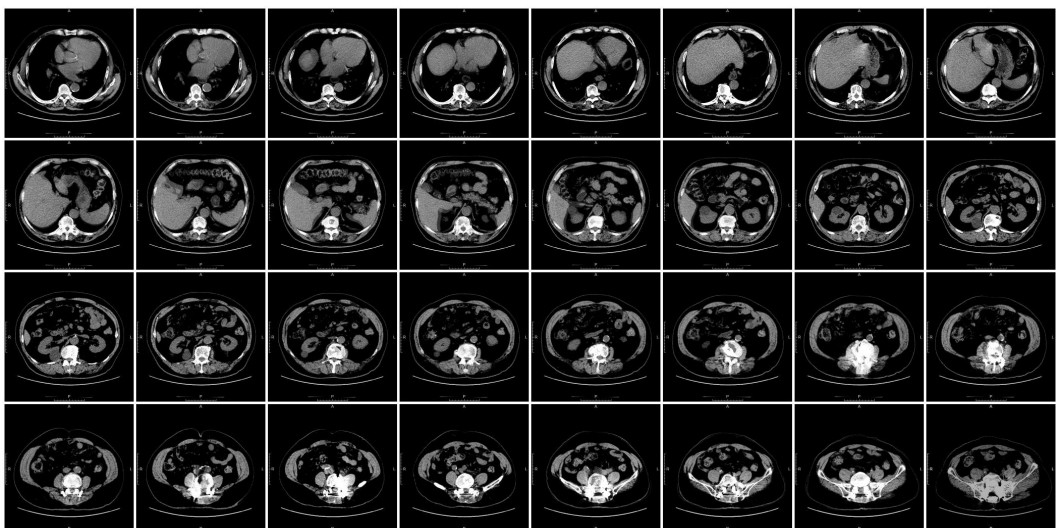

Figure 5: **Non-contrast CT imaging in Axial plane.**

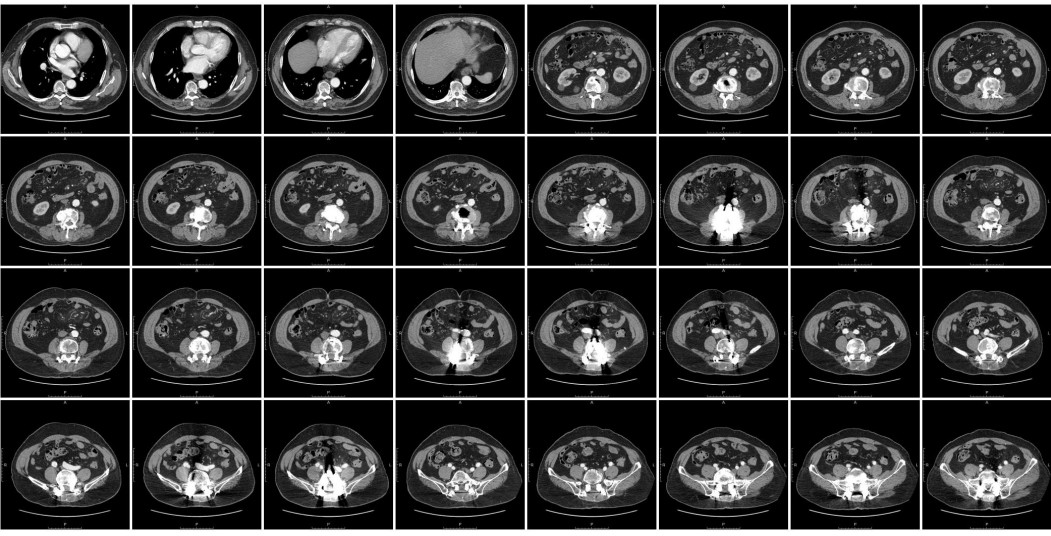

Figure 6: **Arterial CT imaging in Axial plane.**

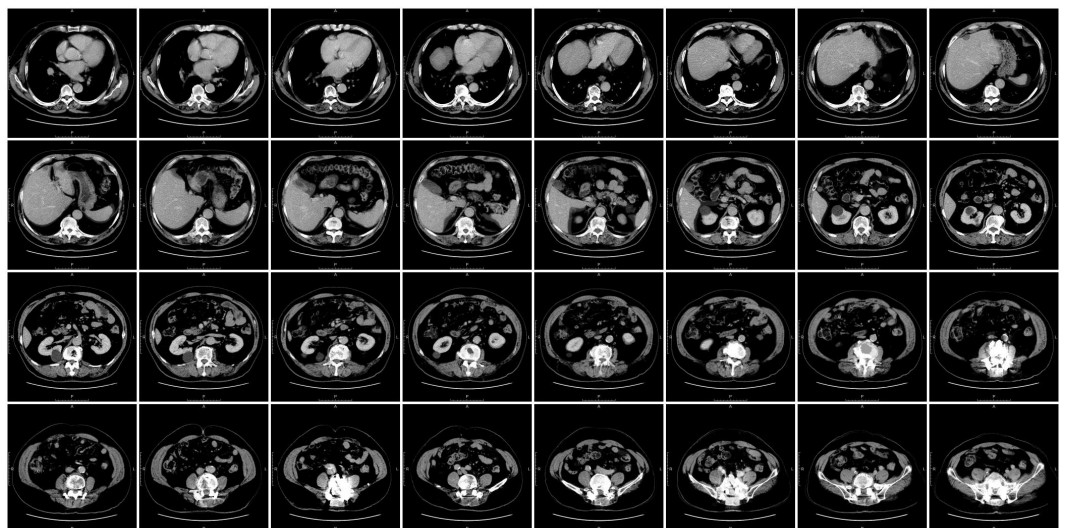

Figure 7: **Portal Venous CT imaging in Axial plane.**

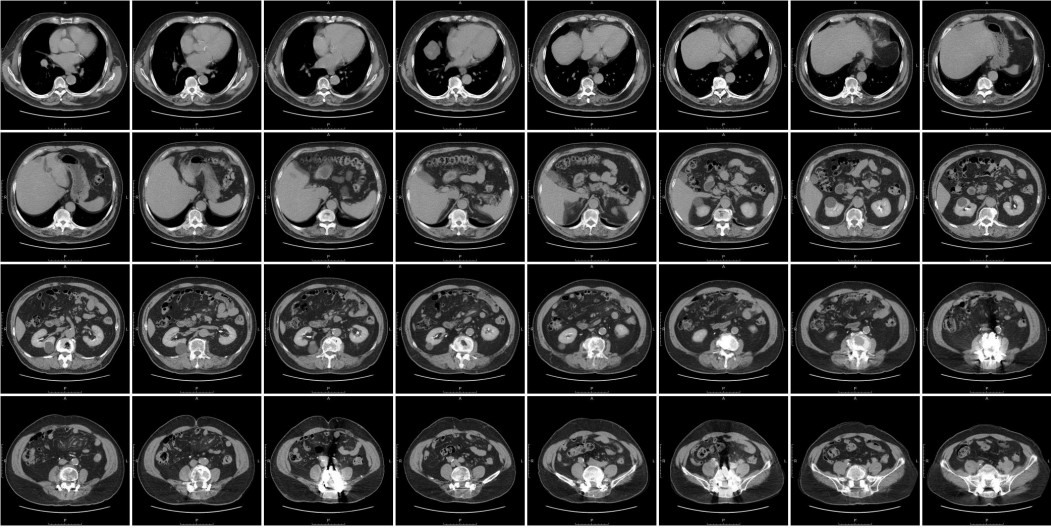

Figure 8: **Delayed CT imaging in Axial plane.**

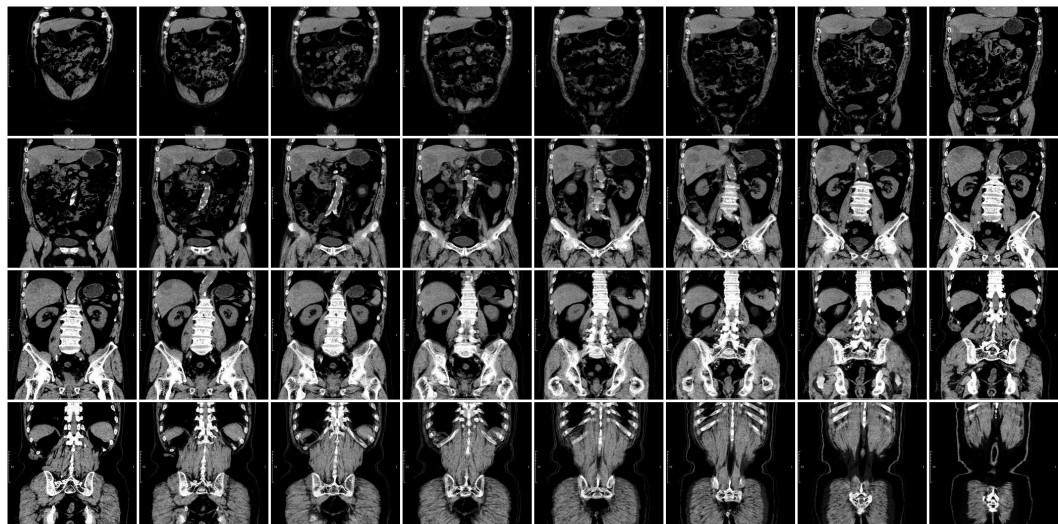

Figure 9: **Non-contrast CT imaging in Coronal plane.**

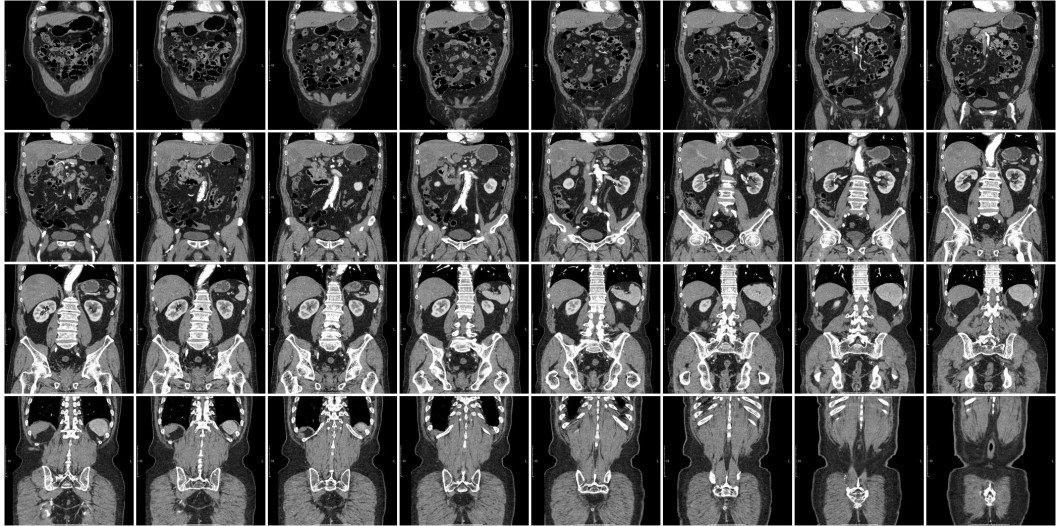

Figure 10: **Arterial CT imaging in Coronal plane.**

**Coronal Plane.**    We demonstrate a phase series acquired in the coronal plane, including non-contrast phase (Figure 9), arterial phase (Figure 10), venous phase (Figure 11) and delayed phase (Figure 12) with the following clinical caption:

*Findings: Contrast-enhanced CT of the abdomen and pelvis with CTA reconstruction shows normal liver contour and density, without biliary dilatation. The gallbladder, spleen, and adrenal glands appear unremarkable. The pancreas shows localized steatosis in the head but no ductal dilation. Both kidneys are normal in shape, with bilateral renal cysts ( 3.3 cm, 6 HU) showing no enhancement. No hydronephrosis or ureteral dilatation is seen. The bowel appears normal, without obstruction or mass. The bladder is underfilled but without wall thickening or intraluminal lesions. The prostate and seminal vesicles are normal. No lymphadenopathy or ascites is observed. Mild degenerative bone changes are noted. CTA reveals abdominal aortic wall calcifications.*

**Sagittal Plane.**    We present an imaging phase series for Sagittal plane, including non-contrast phase (Figure 13), arterial phase (Figure 14), venous phase (Figure 15) and delayed phase (Figure 16) with the following clinical caption:

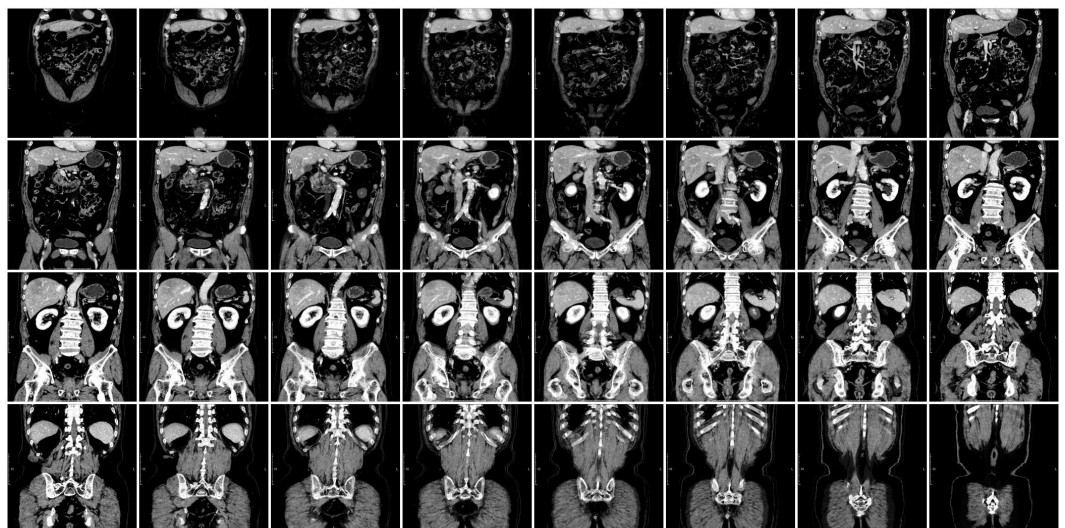

Figure 11: **Portal Venous CT imaging in Coronal plane.**

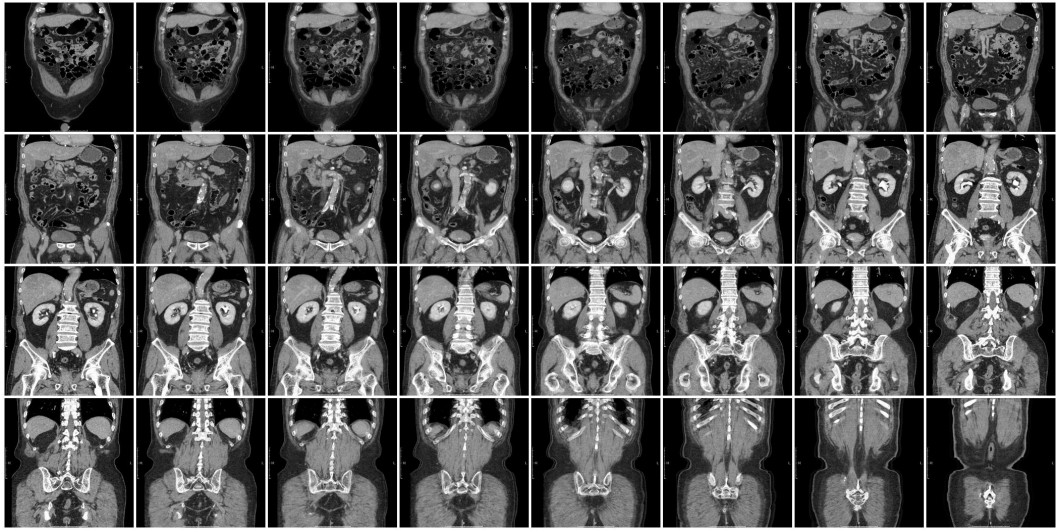

Figure 12: **Delayed CT imaging in Coronal plane.**

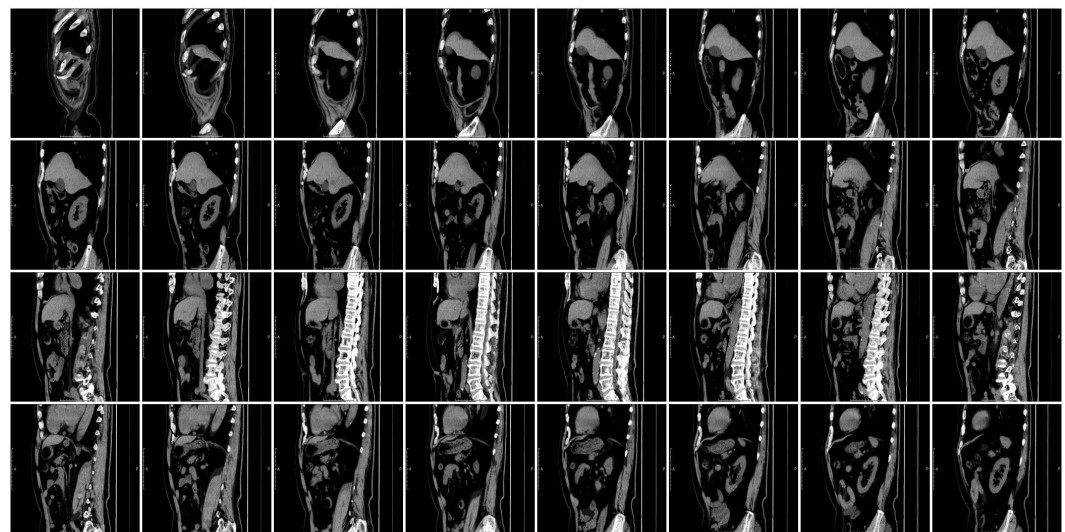

Figure 13: **Non-contrast CT imaging in Sagittal plane.**

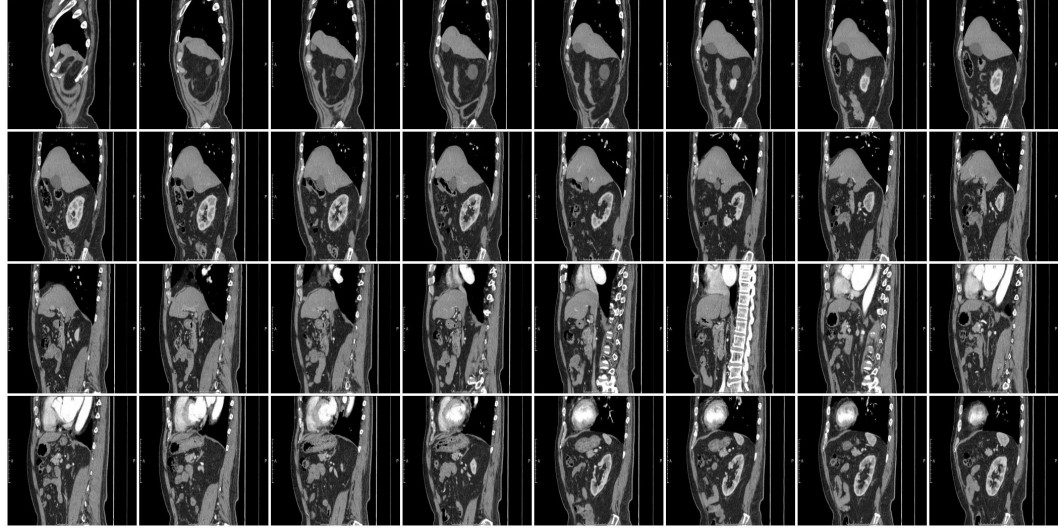

Figure 14: **Arterial CT imaging in Sagittal plane.**

*Findings: The liver is normal in size and contour, with no abnormal parenchymal density or biliary dilatation. The hepatic portal vein is not widened. Gallbladder morphology is normal, without wall thickening or intraluminal lesions. The pancreas is normal in size and shape, showing uneven parenchymal density and localized steatosis in the head, without ductal dilatation. The spleen is unremarkable. Both kidneys are normal in shape, with bilateral low-density renal cysts ( 3.3 cm, 6 HU) showing no enhancement. No ureteral dilation or hydronephrosis is noted. Adrenal glands appear normal. No mass or obstruction is seen in the bowel. The bladder is underfilled but without evident lesions. The prostate and seminal vesicles are unremarkable. Rectal wall and perirectal fat are normal. No abdominal or retroperitoneal lymphadenopathy is identified. Mild degenerative bony changes are present. CTA reveals abdominal aortic and branch calcifications.*

## C    EXPERIMENTAL SETTINGS

All experiments were conducted on an Inspur NF5468M6 server equipped with 8 NVIDIA A100 GPUs, using DeepSpeed for bf16 mixed-precision training to maximize computational efficiency. The

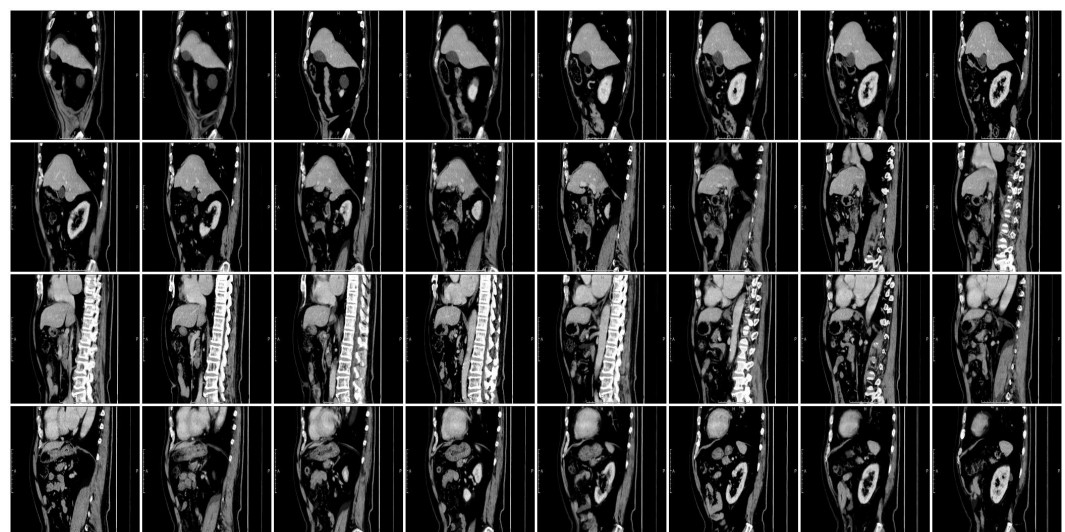

Figure 15: **Portal Venous CT imaging in Sagittal plane.**

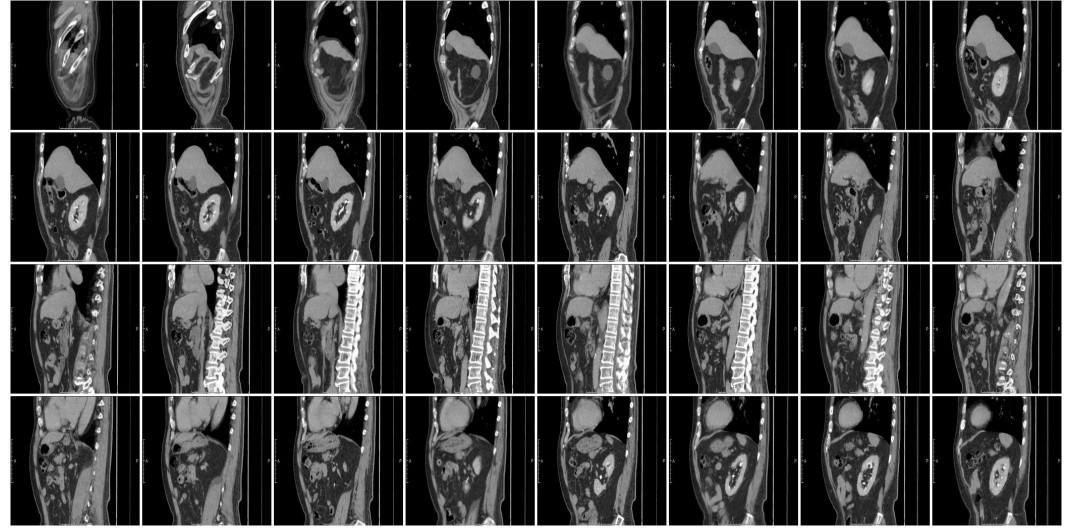

Figure 16: **Delayed CT imaging in Sagittal plane.**

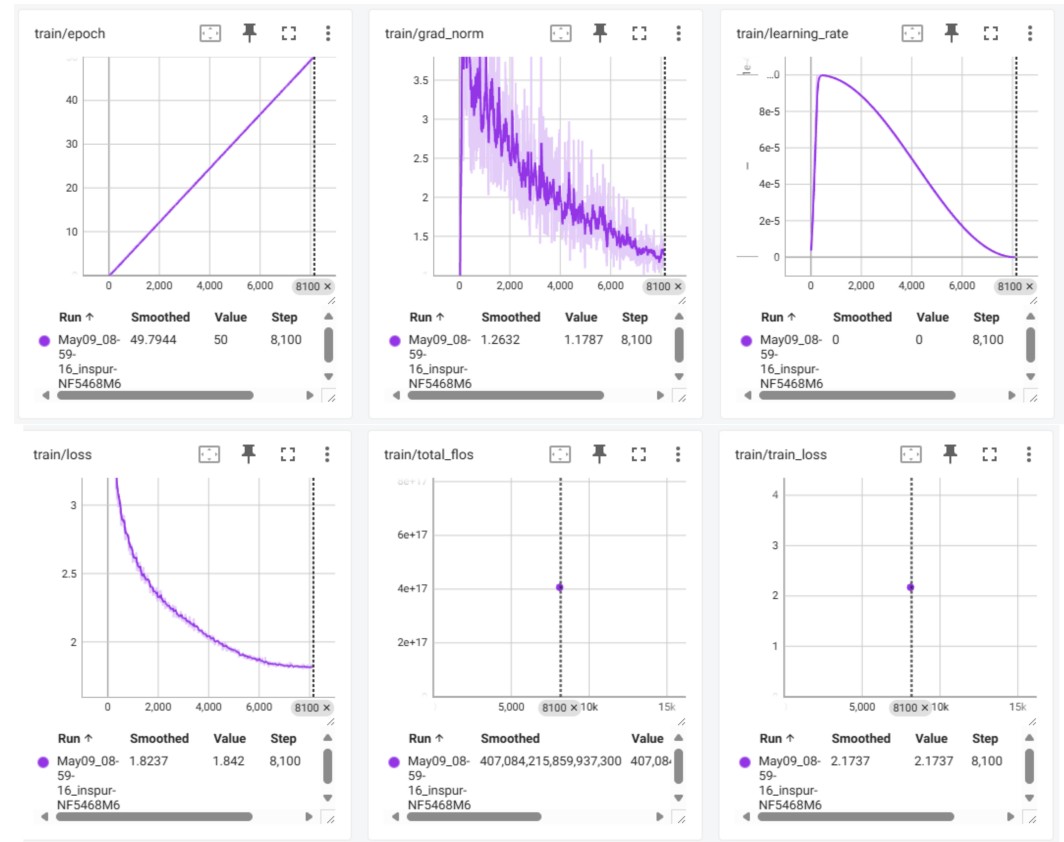

Figure 17: **Pretraining specifics logged by Tensorflow.**

input 3D CT volumes were min–max normalized and resized to $32 \times 256 \times 256$, while corresponding diagnostic texts were tokenized and truncated to 512 tokens. A 12-layer 3D Vision Transformer (ViT) with a patch size of $4 \times 16 \times 16$ was employed to extract spatial representations, producing embeddings of shape $2049 \times 768$, which were then processed alongside the textual tokens using a 12-layer pretrained BERT model. Pretraining was performed with the AdamW optimizer, an initial learning rate of $1 \times 10^{-4}$, a linear warm-up schedule, and cosine decay. As shown in Figure 17, the training phase demonstrated steady loss reduction, declining gradient norms, and a smooth learning rate curve over 8100 steps. For fine-tuning, the parameters of both the visual encoder and the LLaMA-3.1-8B language model were frozen, and optimization was restricted to the lightweight 3D adapter modules. The fine-tuning phase, visualized in Figure 18, shows continued convergence and stable gradient flow. To evaluate downstream performance, multimodal retrieval was assessed using Recallk (R1, R5, R10) for both image-to-text and text-to-image scenarios, while radiology report generation was evaluated using BLEU, ROUGE-1, METEOR, and BERTScore F1 metrics to capture both lexical and semantic quality.

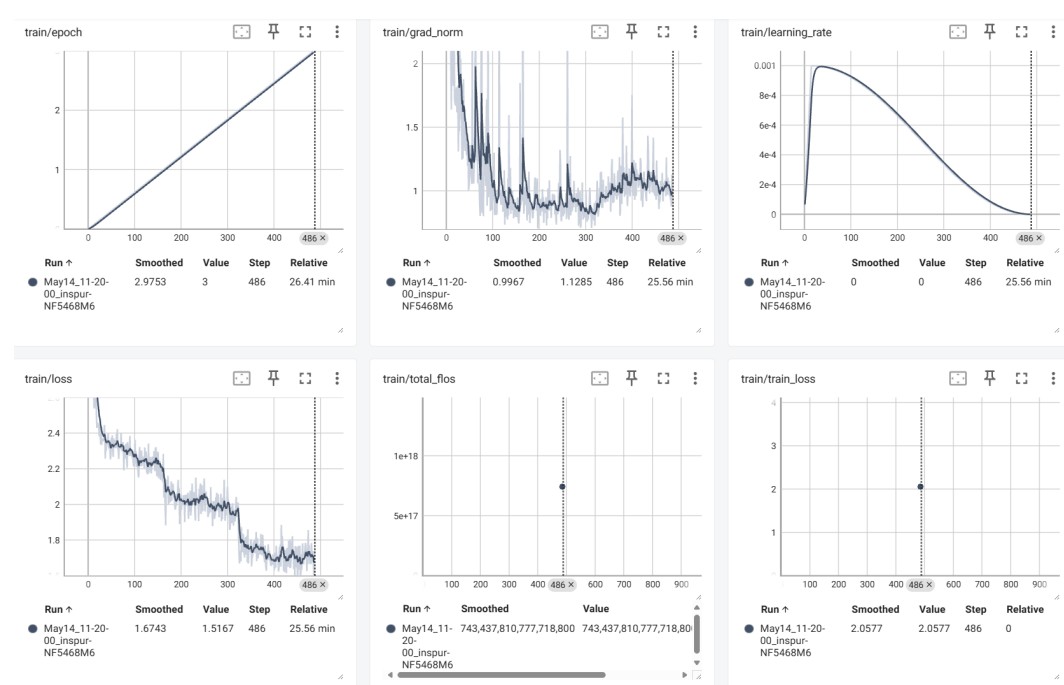

Figure 18: **Tuning specifics logged by Tensorflow.**

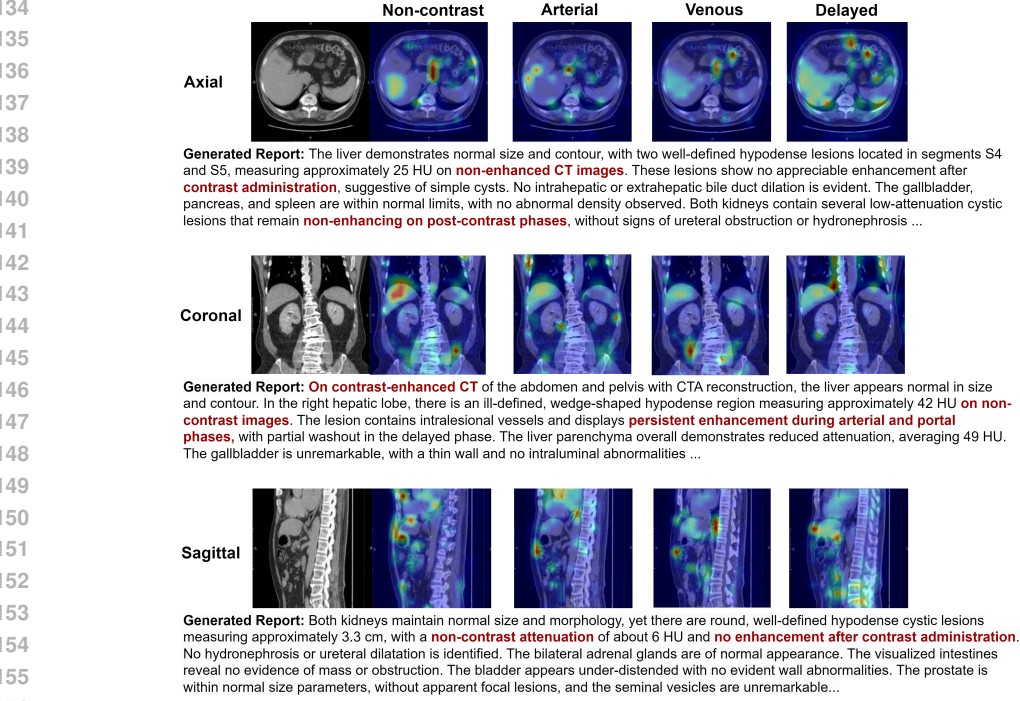

Figure 19: Attention score heatmap overlays across non-contrast and contrast-enhanced CT phases (axial, coronal, sagittal views) highlighting liver and renal lesions described in the generated report.

## D  RESOURCES

We provide an analysis of computational efficiency and scalability of PhoT relative to baselines. Table 6 summarizes GPU memory usage, training/inference time, and scalability. Although PhoT requires moderately higher GPU memory and slightly longer inference per sample, this is directly linked to its strength of modeling multi-phase temporal information. Optimization strategies such as pruning and quantization further reduce resource requirements.

| Model | #Phases | Peak GPU Mem (GB) | Time per Epoch (min) | Inference Time (sec/sample) |
|---|---|---|---|---|
| Vanilla Baseline | 1 (3D) | 15.3 | 11.8 | 1.35 |
| PhoT (Ours) | 2–4 | 19.6 | 17.6 | 2.15 |

Table 6: Resource usage comparison between baseline and PhoT.

## E  CASE STUDY

As shown in Figure 19, the PhoT-generated report effectively integrates clinical information across multiple contrast-enhanced CT phases, accurately describing hypodense liver lesions in segments S4 and S5 ( 25 HU on non-enhanced images) as simple cysts, with no enhancement throughout arterial, venous, and delayed phases. Multiple renal cysts are similarly identified as non-enhancing lesions. Heatmaps across axial, coronal, and sagittal views confirm PhoT's appropriate attention to these lesions during all imaging phases.

# F CODE IMPLEMENTATION

### F.0.1 TRAINING CODE

Pseudo-code for Phase-aware Memory Thought (PhoT) Pretraining:

```
Initialize Vision Transformer (ViT)
Initialize Text Encoder (BERT)
Initialize Contrastive Learning components

For each epoch in epochs:
    For each batch in training data:
        # Load multi-phase imaging data and corresponding diagnostic text
        images, texts = load_batch()

        # Phase-aware feature extraction
        memory_state = initialize_memory()
        For each phase t in phases:
            image_embeddings = ViT(images[t])
            multi_scale_features = multi_scale_convolutions(image_embeddings
                )

            # Update memory state with gated mechanism
            gates = compute_gates(image_embeddings, memory_state)
            memory_candidate = integrate_features(multi_scale_features,
                memory_state, gates)
            memory_state = update_memory(memory_state, memory_candidate,
                gates)

        # Aggregate memory state into compact representation
        visual_embedding = attention_pooling(memory_state)
        normalized_visual_embedding = L2_normalize(project_embedding(
            visual_embedding))

        # Encode textual descriptions
        text_embedding = TextEncoder(texts)
        normalized_text_embedding = L2_normalize(text_embedding)

        # Compute contrastive loss
        similarity_matrix = compute_similarity(normalized_visual_embedding,
            normalized_text_embedding)
        loss = contrastive_loss(similarity_matrix)

        # Backpropagate loss
        optimize_model(loss)

# Save pretrained model and tokenizer
save_model(ViT, TextEncoder, tokenizer)

Pseudo-code for Phase-aware Memory Thought (PhoT) Fine-tuning:

Load Pretrained Vision Encoder (Frozen)
Initialize Spatial Adaptor
Load Language Model (LLM, e.g., LLaMA-3)

For each epoch in epochs:
    For each batch in fine-tuning data:
        # Load multi-phase imaging data and corresponding diagnostic report
        images, reports = load_batch()

        # Extract visual features using frozen pretrained Vision Encoder
        with torch.no_grad():
            memory_state = initialize_memory()
            For each phase t in phases:
                image_embeddings = Pretrained_Vision_Encoder(images[t])
```

```
1242            multi_scale_features = multi_scale_convolutions(
1243                image_embeddings)
1244
1245            # Update memory state with gated mechanism (frozen weights)
1246            gates = compute_gates(image_embeddings, memory_state)
1247            memory_candidate = integrate_features(multi_scale_features,
                    memory_state, gates)
1248            memory_state = update_memory(memory_state, memory_candidate,
1249                gates)
1250
1251        # Spatial adaptor transforms visual memory state into tokens for
              LLM
1252        pooled_memory = spatial_downsample(memory_state)
1253        visual_tokens = linear_projection(pooled_memory)
1254
1255        # Generate textual report with Language Model
1256        predicted_report = LLM_generate(visual_tokens)
1257
1258        # Compute language modeling loss
        loss = language_model_loss(predicted_report, reports)
1259
1260        # Backpropagate loss only through spatial adaptor
1261        optimize_adaptor(loss)
1262
# Save fine-tuned adaptor and tokenizer
1263 save_model(Spatial_Adaptor, tokenizer)
1264
1265
1266 F.0.2 EVALUATION CODE
1267
Pseudo-code for Phase-aware Memory Thought (PhoT) Evaluation:
1268
1269 Load Pretrained Vision Encoder (Frozen)
1270 Load Fine-tuned Spatial Adaptor
1271 Load Fine-tuned Language Model (LLM, e.g., LLaMA-3)
1272
For each sample in evaluation dataset:
1273    # Load multi-phase imaging data and corresponding text
1274    images, ground_truth_text = load_sample()
1275
1276    # Extract visual features using frozen pretrained Vision Encoder
1277    with torch.no_grad():
1278        memory_state = initialize_memory()
        For each phase t in phases:
1279            image_embeddings = Pretrained_Vision_Encoder(images[t])
1280            multi_scale_features = multi_scale_convolutions(image_embeddings
1281                )
1282
            # Update memory state with gated mechanism (frozen weights)
1283            gates = compute_gates(image_embeddings, memory_state)
1284            memory_candidate = integrate_features(multi_scale_features,
1285                memory_state, gates)
1286            memory_state = update_memory(memory_state, memory_candidate,
1287                gates)
1288    # Spatial adaptor transforms visual memory state into tokens for LLM
1289    pooled_memory = spatial_downsample(memory_state)
1290    visual_tokens = linear_projection(pooled_memory)
1291
1292    # Generate text prediction with Language Model
    predicted_text = LLM_generate(visual_tokens)
1293
1294    # Compute retrieval metrics
1295    Compute similarity between visual_tokens and ground_truth_text
        embeddings
```

```
    Calculate retrieval metrics (e.g., Recall@1, Recall@5, Recall@10)

    # Compute generation metrics
    Evaluate predicted_text against ground_truth_text using:
        - BLEU
        - ROUGE-1
        - METEOR
        - BERTScore

Aggregate and report mean metrics for retrieval and generation tasks

Pseudo-code for Phase-aware Memory Thought (PhoT) Inference:

Define caption templates list with various prompts:
    Caption_templates = [
        "Can you provide a caption consisting of findings for this medical
            image?",
        "Describe the findings of the medical image you see.",
        ... (other prompts) ...
        "Can you provide a diagnosis based on this image?"
    ]

Define prompt generation function:
    Function generate_llama3_prompt(original_question):
        system_directive = (
            "You are a medical imaging AI assistant. Your task is to analyze
                the provided medical image "
            "and generate a single, compact paragraph summarizing the
                definitive medical findings. "
            "Focus on accuracy and stick to observable details. "
            "Consider all relevant aspects of the image (e.g., different
                phases if applicable) "
            "to form your synthesized conclusion."
        )
        prompt = system_directive + "\n\nImage Analysis Task: " +
            original_question + "\n\nCaption:"
        Return prompt

For each inference request:
    Select a random caption template from Caption_templates
    Generate the prompt using generate_llama3_prompt(selected_template)

    Load multi-phase imaging data
    Extract visual tokens using frozen pretrained Vision Encoder and
        Spatial Adaptor (as previously defined)

    Generate medical caption using Language Model (LLM) based on visual
        tokens and generated prompt

    Return generated medical report or caption
```

