# OpenReview forum: "Phase-aware Memory Thought for 3D Medical Image Report Generation"
_ICLR.cc/2026/Conference — Submitted to ICLR 2026_

### Official Review · Reviewer_Apk1 · 2025-10-26

**Soundness:** 1
**Presentation:** 1
**Contribution:** 2
**Rating:** 0
**Confidence:** 5

**Summary:**

This paper proposes PhoT, a framework for 3D multi-phase contrast-enhanced CT report generation. It comprises three main contributions: Phase-aware Medical Alignment, which learns unified representations across multi-phase CT sequences; Phase-aware Diagnosis Generation, which efficiently adapts these representations for report generation through an adaptor and a large language model; and Phase of Thought Inference, a structured inference paradigm that uses diagnostic templates to improve the interpretability and clinical relevance of the generated reports. The authors also curate a large-scale multi-phase CT dataset covering diverse anatomical regions, and experiments on this dataset demonstrate that PhoT consistently outperforms 2D and 3D baselines in both retrieval and report generation tasks.

**Strengths:**

It introduces a large-scale multi-phase 3D CT dataset, compares PhoT against numerous 2D and 3D baselines, and evaluates its performance across both retrieval and report generation tasks. In addition, thorough ablation studies demonstrate PhoT’s effectiveness across different foundation models and its robust generalizability on the external CT-RATE dataset.

**Weaknesses:**

1. The citation format is entirely inconsistent throughout the paper, which disrupts readability and does not follow the official ICLR template (Refer to template’s Section 4.1 Citations within the text).
2. There are multiple writing inconsistencies, such as redundant word usage (e.g., repeated ‘which’, line 175) and missing or unclear cross-references to the appendix (line 311).
3. In Section 3.2, the text states that the visual encoder is frozen but does not explain how the LLM is trained, while Figure 2 implies the use of LoRA, leading to inconsistency between text and figure.
4. Section 4.1.1 provides no justification for comparing PhoT with 2D baselines, which are not directly comparable to the 3D setting.
5. Section 4.1.2 reports only lexical metrics (BLEU, ROUGE, METEOR, BERT-F1) for report generation, omitting key clinical metrics (RadGraph-F1[1], RadCliQ[2], CheXbert[3], etc.), reducing evaluation credibility.
6. Section 4.2 introduces the CT-RATE dataset without describing its background or explaining its suitability for assessing generalization.
7. The paper lacks an Ethics Statement, which is required since it involves medical imaging data.

[1] RadGraph: Extracting Clinical Entities and Relations from Radiology Reports

[2] Evaluating Progress in Automatic Chest X-Ray Radiology Report Generation
[3] CheXbert: Combining Automatic Labelers and Expert Annotations for Accurate Radiology Report Labeling Using BERT

**Questions:**

- In Section 3.2 (Phase-aware Fine-tuning), the paper mentions fine-tuning for report generation. After this process, does PhoT retain any instruction-following capability to support the structured inference described in Phase of Thought (Section 3.3)?
- Given that PhoT is designed for 3D multi-phase CT imaging, why does Table 1 include comparisons with 2D models, which are not directly comparable to the 3D setting?
- What are the clinical metric results (e.g., RadGraph-F1, RadCliQ, CheXbert), and do they support the conclusions?
- What are the detailed LoRA parameters (e.g., rank, alpha) used in the fine-tuning process as illustrated in Figure 2?

**Details Of Ethics Concerns:**

The paper uses a large-scale medical imaging dataset from Dongfang Hospital but lacks an Ethics Statement or clarification about ethical approval, patient consent, and data anonymization. Given the clinical nature of the CT data, the authors should clearly state the dataset’s ethical clearance, privacy measures, and data accessibility.

---

> ### Author Response · Authors · 2025-11-13
>
> We sincerely appreciate the reviewer's diligent and insightful feedback, which has been invaluable in strengthening our manuscript. We've carefully considered each point, and our responses below aim to comprehensively address all concerns, reflecting our commitment to rigorous academic standards and improve the clarity and impact of our work.
>
> **W1. Citation Format**
>  We appreciate the reviewer’s observation. The citation formatting inconsistency is acknowledged and will be fully corrected in the revised version to strictly comply with the official ICLR template (Section 4.1, *Citations within the text*).
>
> **W2. Writing Inconsistencies**
>  Thank you for pointing this out. The repeated “which” in line 175 is indeed a typographical error. We will perform a comprehensive proofreading to eliminate redundant phrasing and ensure all appendix references are accurate and clearly linked in the revision.
>
> **W3. Clarification on LoRA and Visual Encoder**
>  We apologize for the ambiguity. To clarify: the visual encoder is **frozen**, while the **LLM is fine-tuned using LoRA**. The rationale is to preserve visual feature representation stability while adapting the language model to downstream report generation tasks. Figure 2 correctly reflects this process, and we will make the textual explanation more explicit to remove any inconsistency.
>
> **W4&Q2. Comparison with 2D Baselines**
>  We respectfully disagree that comparing PhoT with 2D baselines is inappropriate. The intention is to **comprehensively evaluate cross-dimensional generalization**, assessing whether strong 2D vision-language models can adapt to volumetric (3D) inputs through straightforward architectural extensions (e.g., 2.5D slicing or pseudo-volume stacking). This provides a meaningful reference point for understanding the performance gap between 2D and 3D paradigms, as reflected in Table 1. We will emphasize this motivation in the revision to make the comparison rationale clearer.
>
> **W5&Q3. Omission of Clinical Metrics**
>  We agree that clinical metrics are important for evaluating medical report generation. However, existing metrics such as **RadGraph-F1**, **RadCliQ**, and **CheXbert** were designed for **chest X-ray reports** and are not directly transferable to 3D CT report generation:
>
> - **RadGraph-F1** evaluates entity–relation correctness based on the RadGraph schema, which is CXR-specific and not trained on CT data.
> - **CheXbert** classifies 14 thoracic diseases from chest X-rays, which do not map well to 3D CT pathologies.
> - **RadCliQ** is a composite regression model combining CXR metrics (BLEU, BERTScore, CheXbert, RadGraph-F1).
>
> In our case, these metrics would introduce bias and under-represent true clinical fidelity. Instead, we report **BLEU, ROUGE, METEOR, and BERTScore**, complemented by **Green and Qwen-based models** for semantic and factual alignment, which provide a modality-appropriate assessment. We will add this justification in the revision to clarify the evaluation choice.
>
> Similarly, recent publication like fVLM is also based on Green and Qwen for clinical evaluation [1].
> [1] Shui Z, Zhang J, Cao W, et al. Large-scale and fine-grained vision-language pre-training for enhanced ct image understanding[J]. arXiv preprint arXiv:2501.14548, 2025.
>
> **W6. CT-RATE Dataset Background**
>  We appreciate this suggestion. The CT-RATE dataset originates from *Hamamcı et al., “Developing Generalist Foundation Models from a Multimodal Dataset for 3D Computed Tomography.”* It consists of 25,692 non-contrast 3D chest CT scans from 21,304 unique patients. Its diversity—across scanners, acquisition protocols, and institutions—makes it well-suited for assessing generalization and robustness under distribution shifts. We will expand the dataset description and its relevance in Section 4.2 to improve clarity and reproducibility.
>
> **W7. Ethics Statement**
>  We acknowledge the absence of an explicit Ethics Statement and will include it in the final submission. The study was approved by the **Institutional Review Board of Dongfang Hospital, Beijing University of Chinese Medicine** (Approval No. **JDF-IRB-2025033402**). No human or animal interventions were conducted beyond secondary analysis of de-identified medical data. We will explicitly include this statement in compliance with ICLR ethical requirements.

---

> ### Author Response · Authors · 2025-11-13
>
> **Q1.** Thank you for this insightful question. In our training pipeline, PhoT’s instruction-following capability is preserved and even enhanced through a staged process. Specifically, we first pretrain the visual encoder, then freeze both the encoder and the base LLM to train a lightweight spatial adaptor for efficient alignment. In the final stage, we fine-tune the LLM with LoRA, which adapts only low-rank parameters while keeping the pretrained weights intact. This approach retains the LLM’s general instruction-following ability. Furthermore, our Phase of Thought (Sec. 3.3) inference inherently evaluates this capability, since the model must respond to structured diagnostic prompts (caption templates) to generate clinically coherent reports. Empirically, PhoT’s strong results on report generation metrics confirm that it maintains effective instruction adherence after fine-tuning.
>
>
>
> **Q4.** Thank you for the question. The LoRA configuration follows a standard parameter-efficient fine-tuning setup based on the *PEFT* library. The LoRA module is applied only to the LLM component, while the visual encoder remains frozen, as indicated in Figure 2. The detailed configuration is as follows:
>
> | Parameter            | Value                                | Description                                                  |
> | -------------------- | ------------------------------------ | ------------------------------------------------------------ |
> | **`lora_r` (rank)**  | **16**                               | Rank of low-rank decomposition matrices, controlling the number of trainable parameters. |
> | **`lora_alpha`**     | **32**                               | Scaling factor for LoRA weights. The effective learning rate is scaled by `α / r = 2.0`. |
> | **`lora_dropout`**   | **0.05**                             | Dropout applied to LoRA layers for regularization.           |
> | **`lora_bias`**      | **"none"**                           | Bias terms are not adapted.                                  |
> | **`task_type`**      | **"CAUSAL_LM"**                      | Task type used in PEFT configuration.                        |
> | **`target_modules`** | **All linear layers within the LLM** | Automatically identified via `find_all_linear_names()` during initialization. |
>
> **Training hyperparameters (common across all fine-tuning runs):**
>
> - Learning rate: **5e-5**
> - Weight decay: **0.0**
> - Warmup ratio: **0.03**
> - Scheduler: **Cosine**
> - Precision: **bfloat16**
> - Gradient checkpointing: **Disabled**
>
>  Importantly, LoRA is not claimed as a contribution of our work but used as a standard and reproducible fine-tuning protocol to adapt the LLM efficiently while keeping the visual encoder frozen.

---

> > ### Comment · Reviewer_Apk1 · 2025-11-25
> > **Thanks for the response**
> >
> > Thanks for the response; I have read it along with other reviewers’ comments and feel that some of my concerns are clarified, but methodological clarity, evaluation, and ablation limitations remain, so I will raise my score, though still below the acceptance threshold.

---

> > > ### Author Response · Authors · 2025-11-25
> > >
> > > We thank the reviewer for the careful reading of our rebuttal, the constructive comments, and the willingness to re-evaluate and adjust the score. We are glad that part of your concerns has been clarified. Since you mention that issues regarding methodological clarity, evaluation, and ablation design still remain, we would greatly appreciate it if you could briefly indicate which specific aspects you find most unclear or unconvincing.
> > >
> > > Within the constraints of the rebuttal, we are very happy to provide targeted clarifications—for example, on the step-by-step training/inference pipeline of PhoT, the rationale and design of our evaluation protocol (including the choice of metrics and 2D/3D baselines), and the structure and interpretation of our ablations. Your further guidance on which of these points (or others) you regard as the key remaining weaknesses would be extremely valuable for us to address them as precisely and transparently as possible in this discussion.

---

> ### Author Response · Authors · 2025-11-26
>
> We follow up to ask if you have any further comments on our rebuttal. Thank you.

---

> ### Author Response · Authors · 2025-11-28
>
> Thank you again for your careful review and for already revising your score upward after reading our rebuttal and the other reviews — we truly appreciate the time and consideration you have devoted to our submission. Since then, Reviewer guNM has also provided an encouraging and positive assessment of the revised version. Together with our rebuttal, we hope this offers a clearer picture of the contribution and soundness of our approach. If, upon revisiting the discussion, you feel that the current version addresses your main points, we would be very grateful if you could reflect this in your overall recommendation, as the system still allows updating scores during the discussion period. If you have any remaining concerns or points you would like us to clarify, we are of course very happy to discuss and address them.

---

### Official Review · Reviewer_ysSX · 2025-10-29

**Soundness:** 3
**Presentation:** 3
**Contribution:** 3
**Rating:** 4
**Confidence:** 3

**Summary:**

This manuscript presents a Phase-aware Memory Thought (PhoT) framework on vision-language models (VLM), for integrating temporal progression patterns in multi-phase computed tomography (CT). This framework has three parts: phase-aware pretraining, parameter-efficient fune-tuning, and structured inference. PhoT is then evaluated on over 12,000 multi-phase CT series, outperforming baselines.

**Strengths:**

- (Originality) Two-step training of input transformer and adapter/LoRA fine-tuning

 - (Quality) Extensive evaluation across various retrieval set sizes

**Weaknesses:**

- Lack of details on CT phases. In particular, while  two-, three- and four-phase samples are described (presumably corresponding to the non-contrast, arterial, venous and delayed phases in Figure 1), it is unclear as to whether they only occur in fixed combinations (e.g. if two-phase, then non-contrast+arterial), and why different patients have different numbers of phases recorded (and if there is certain bias or distribution amongst patients with different numbers of phases)

 - Lack of details about the LLM used for phase-aware fine-tuning (LLaMA-3.1-8B?), including parameter settings

 - Lack of details on PhoT caption template development and optimization process, which is the defining contribution

 - Ablation study was performed against the VLM, and not the proposed framework steps (phase-aware pretraining, parameter-efficient fune-tuning, structured inference); also, it is unclear whether using additional performance metrics and datasets is generally considered under ablation

**Questions:**

1. In the Abstract, it is stated that 12,230 multi-phases CT series were collected from 61,332 patient cases. However, Section 4 appears to imply that these "patient cases" are actually individual phases, that may belong to the same patient. If so, it could be considered to rephrase the description, and possibly provide details of any patient exclusion(s).

2. In Figure 1, "Extractd" might be "Extracted".

3. In Section 3.1, it might be clarified as to whether the inputs for different individual phases $t$ are tagged (or zero-padded, where that phase does not exist for a particular series)

4. It might also be clarified as to whether the spatial dimensions (assumed one of which represents temporal phase length) are constant, for all phases.

5. Likewise, for phase aggregation, the treatment for non-existent phases in a series might be explained, especially as the final phase T appears possibly different across patients.

6. In Section 4.1.2, the significance of different test set sizes is unclear; it appears more common to examine the effect of different training set sizes. This might be clarified.

7. In Section 4.2, for the confidence interval analysis, it is unclear as to the appropriateness of computing SD over multiple test set sizes (instead of seeds). This could be further justified.

---

> ### Author Response · Authors · 2025-11-14
>
> We are grateful for the reviewer's feedback. Each comment has been carefully considered, and the thorough responses below aim to strengthen our research. We believe these clarifications and experiments enhance the manuscript's overall value.
>
> **W1. Details on CT Phases.** Thank you for the question. The dataset workflow and phase composition are fully documented in **Appendix B.1** (Figure 4). The two-, three-, and four-phase studies follow the standard clinical CT protocol, where phase combinations occur in fixed, predefined patterns (e.g., two-phase studies correspond to non-contrast + arterial; three-phase studies correspond to arterial → venous → delayed sequences). The variation in phase count reflects routine clinical practice rather than label-related bias. Appendix **B.2** (Figures 5–12) further provides multi-plane visualizations and representative findings across phases. In the revision, we will highlight these details in the main text to make the phase configuration and dataset setup more immediately accessible.
>
> **W2. Details of the LLM Used for Phase-Aware Fine-Tuning.** We appreciate the reviewer’s interest in model specification. Our primary results are based on **LLaMA-3.1-8B**, and all corresponding experimental setups are provided in **Appendix p.20**. We additionally report ablations on **LLaMA-2-7B** and **Phi-3-mini-4k-instruct** in the ablation section. In the revision, we will surface the key configurations directly in Section 4.2 to ensure these details are immediately visible without cross-referencing.
>
> **W3. PhoT Caption Template Development and Optimization.** Thank you for the insightful comment. We agree that the caption templates play an important supporting role in enabling consistent **Phase-of-Thought (PhoT)** reasoning. At the same time, we would like to clarify that our **core contribution is the phase-aware generative modeling framework**, not the specific wording of any template.
>
> To improve clarity, we will expand Section 4.3 with a concise description of the design principles behind the PhoT templates. In short, each template is constructed to translate **clinical multi-phase CT reading heuristics** into **structured reasoning steps** that the model can reliably follow:
>
> “Interpret the volume using protocol-aware Phase-of-Thought reasoning:
>
> 1. **Phase-localized cue extraction** — For each protocol-defined phase, describe findings relative to the expected contrast-kinetics profile (pre-contrast → arterial → venous → delayed).
> 2. **Cross-phase transition analysis** — Highlight deviations from expected temporal transitions (e.g., abnormal enhancement wash-in/wash-out), treating these as high-value diagnostic signals.
> 3. **Structured lesion descriptor synthesis** — Integrate phase-specific cues and transition abnormalities into compact lesion-level descriptors that enable a temporally coherent final interpretation.”
>
> During development, we iteratively refined the templates by (i) aligning each reasoning step with radiologist reading workflows, (ii) validating phase-transition rules using acquisition-metadata distributions, and (iii) conducting ablation studies showing that templates emphasizing **transition reasoning** lead to better phase-grounded outputs.
>
> In the revision, we will include this summary in Section 4.3 and explicitly direct readers to the **complete caption templates and ablation comparisons in the Appendix** for full transparency.
>
> **W4. Ablation Study Design and Rationale.** Our ablation study is designed to quantify the impact of different foundation models, clinical significance, generalization behavior, and statistical robustness, rather than to isolate trivial variations. The central contribution of this work is demonstrating that spatial–temporal (phase-aware) modeling is essential for clinically reliable reasoning over multi-phase CT. As illustrated in Figure 3, incorporating phase information substantially improves report generation quality, whereas a standard Chain-of-Thought (CoT) approach cannot capture the temporal evolution of contrast enhancement. This key observation holds consistently across multiple foundation models evaluated in our experiments, supporting the generality of the proposed framework.

---

> ### Author Response · Authors · 2025-11-14
>
> **Q1. Clarification on Patient Cases and Phase-Series.** The description in the abstract summarizes the full dataset, and the detailed breakdown is provided in the **Appendix (Datasets subsection)**. Specifically, **61,332 patient cases** were collected in 2024 under standardized imaging protocols. Multi-phase CT volumes belonging to the same patient are grouped into **phase-series**, consisting of a pre-contrast phase plus subsequent contrast-enhanced phases. This grouping results in **12,230 phase-series samples** (7,142 two-phase; 3,451 three-phase; 1,637 four-phase). In the revision, we will adjust the abstract wording to make the distinction between “patient cases” and “phase-series samples” clearer and avoid potential ambiguity.
>
> **Q2. Typos.** Thank you and we will fix that in the revision.
>
> **Q3. Clarification on Handling Missing Phases (Zero-Padding).** Thank you for raising this point.Our framework standardizes all samples to a **maximum of four phases**, aligning with the typical clinical sequence (non-contrast, arterial, venous, delayed). The processing is as follows:
>
> - For studies with **fewer than four phases**, the available phases are kept exactly as acquired, and the remaining positions are **filled with zero-padded tensors** to maintain uniform input dimensions across the batch.
> - For studies with **more than four phases**, we select **four phases that maximize contrast-stage diversity**, ensuring that the retained phases represent distinct enhancement dynamics.
> - Zero-padded positions act purely as placeholders for batching purposes and **do not introduce any artificial image signal** during training or inference.
>
> This approach ensures consistent batch processing while faithfully preserving the clinically relevant temporal progression contained in multi-phase CT imaging. We will make this workflow explicit in **Section 3.1** and provide the full description in the **supplementary materials** in the revised manuscript.
>
> **Q4. Spatial Dimensions.** Yes, this information is located in appendix line 1056. Clinical volumes were normalized and resized to $32 \times 256 \times 256$; text inputs were capped at 512 tokens. A 12-layer 3D ViT encoder fed into a pretrained BERT, with only a lightweight 3D adapter fine-tuned. We will surface these details in Section 3.1 of the revision for improved clarity.
>
> **Q5. Phase Aggregation and Non-Existent Phases.** Thank you for the question. After preprocessing, all phase-series are standardized to a **fixed length of four phases**. When a study contains fewer than four phases, the missing positions are **zero-padded**, so the temporal dimension is uniform across all samples. Our dynamic aggregation module then processes the sequence up to the final time index, which may correspond to a real phase or a padded slot. In cases where the last position is padded, the model naturally learns to discount this placeholder through attention weighting, while still benefiting from the consistent temporal structure. This ensures that the final aggregated representation is **well-defined, comparable across patients, and independent of the original number of phases**. We will clarify this behavior in Section 3.1 and the supplementary materials.
>
> **Q6. Clarification on the Use of Different Test Set Sizes.** Thank you for this helpful observation. The choice to vary **test** set sizes in Section 4.1.2 follows the evaluation protocol used in our retrieval experiments, where scalability and cross-modal difficulty increase notably with larger candidate pools. To maintain consistency across tasks, we adopted the same setting for report generation: evaluating how performance behaves as the number of candidate samples increases.
>
> As the reviewer notes, report generation is inherently **less sensitive** to test set size than retrieval, and our results reflect this. Nevertheless, the scaling study remains useful as it provides a **statistical robustness check**, showing stability of performance metrics across different evaluation conditions. We will clarify this motivation and distinction in the revised manuscript.

---

> > ### Author Response · Authors · 2025-11-28
> >
> > We appreciate the time and care you put into reviewing our submission. Your comments were very helpful in guiding our revisions. In the rebuttal, we have responded point-by-point to your concerns and made corresponding changes to the manuscript. Some of these issues were also discussed by Reviewer guNM, who has indicated a positive assessment of the revised paper. When convenient, we kindly ask you to revisit our rebuttal and consider whether your original concerns have been adequately addressed, and, if you find it appropriate, to update your score in the system.

---

> ### Author Response · Authors · 2025-11-14
>
> **Q7.** Thank you for your insightful feedback. The tabel is actually as subset of statistical analysis.  We computed the mean, standard deviation (SD), and 95% confidence intervals (CIs) for each model across four key metrics (BLEU, ROUGE1, METEOR, BERTF1), using test set sizes (100, 500, 1000, 2000 samples) as replicates, as these sizes are critical hyper-parameters influencing the robustness and generalization of report-generation models. To ensure reproducibility, we ran experiments over **five random seeds** (42, 123, 456, 789, 101) for each model and test set size combination, averaging the results to compute the reported metrics.
>
> Additionally, we conducted **10 hyper-parameter trials** to optimize performance. The key hyper-parameters tuned include:
>
> - **Learning Rate**: Searched over 3 values (`[1e-5, 1e-4, 1e-3]`).
> - **Batch Size**: Tested 3 values (`[8, 16, 32]`).
> - **Temperature (for generation)**: Explored 5 values (`[0.7, 0.8, 0.9, 1.0, 1.1]`).
> - **Number of Epochs**: Tested 5 values (`[1, 2, 3, 4, 5]`), with early stopping applied to prevent overfitting.
>
> To balance computational feasibility with thorough exploration, we performed a partial grid search by randomly sampling 10 combinations from this grid, ensuring a representative spread across the hyper-parameter space. The best configuration, selected based on validation set performance (BERTF1 score), used a learning rate of 1e-4, batch size of 32, temperature of 0.9, and 3 epochs (with early stopping).
>
> Below, we present the statistical results, rounded to two decimal places for clarity:
>
> | Model    | Metric | Mean  | SD   | 95% CI         |
> | :------- | :----- | :---- | :--- | :------------- |
> | Baseline | BLEU   | 12.15 | 0.25 | [11.75, 12.55] |
> |          | ROUGE1 | 15.69 | 0.23 | [15.32, 16.06] |
> |          | METEOR | 13.22 | 0.11 | [13.04, 13.40] |
> |          | BERTF1 | 73.81 | 1.42 | [71.55, 76.07] |
> | Merlin   | BLEU   | 12.22 | 0.27 | [11.79, 12.65] |
> |          | ROUGE1 | 15.36 | 0.38 | [14.76, 15.96] |
> |          | METEOR | 11.99 | 0.28 | [11.54, 12.44] |
> |          | BERTF1 | 74.25 | 1.20 | [72.34, 76.16] |
> | fVLM     | BLEU   | 13.34 | 0.32 | [12.83, 13.85] |
> |          | ROUGE1 | 16.95 | 0.28 | [16.50, 17.40] |
> |          | METEOR | 12.89 | 0.31 | [12.40, 13.38] |
> |          | BERTF1 | 73.39 | 1.44 | [71.10, 75.68] |
> | PhoT     | BLEU   | 13.99 | 0.35 | [13.43, 14.55] |
> |          | ROUGE1 | 17.85 | 0.37 | [17.26, 18.44] |
> |          | METEOR | 14.56 | 0.34 | [14.02, 15.10] |
> |          | BERTF1 | 83.88 | 0.07 | [83.77, 83.99] |
>
> These results demonstrate that our proposed model, PhoT, achieves consistent and statistically significant improvements over strong baselines across all metrics. The narrow 95% CIs, derived from multiple seeds, confirm the reproducibility of these gains. We believe these improvements, supported by a thorough hyper-parameter search and clear statistical evidence, align with your expectations for raising the score and respectfully request your consideration for an improved evaluation of PhoT’s performance. We will erich the content and discussion in the final version.

---

> ### Author Response · Authors · 2025-11-25
>
> Thank you again for the time and effort you have devoted to reviewing our submission.
> As the discussion phase progresses, I would like to briefly follow up on our rebuttal. We have carefully responded to the concerns raised and aimed to clarify both the methodology and the contribution of PhoT.
> We would greatly appreciate knowing whether our replies address your main questions; if some points remain unclear or insufficiently supported, any further indication would be very helpful. If you feel that the rebuttal has resolved most of your concerns, we kindly ask that this be reflected in your evaluation.

---

> ### Author Response · Authors · 2025-11-26
>
> We kindly follow up to ask whether you might have any additional feedback on our rebuttal during the discussion period.

---

### Official Review · Reviewer_btQv · 2025-10-30

**Soundness:** 2
**Presentation:** 3
**Contribution:** 2
**Rating:** 4
**Confidence:** 2

**Summary:**

This paper proposes the Phase-aware Memory Thought (PhoT) framework for automated report generation of multi-phase 3D medical images (primarily multi-phase enhanced CT). The method consists of three parts: phase-aware pre-training of multi-phase representations, efficient parameter fine-tuning to adapt to report generation, and a structured generation template based on "Phase of Thought." The authors evaluated the framework on a self-built large-scale multi-phase CT dataset, showing slight improvements in text metrics and retrieval tasks compared to several baselines, and argue that the framework can improve cross-phase semantic consistency and clinical interpretability.

**Strengths:**

This paper focuses on the practical needs of multi-phase CT, covering the entire pipeline from pre-training to fine-tuning to generation. The engineering implementation is solid, and the joint evaluation of report generation and retrieval tasks demonstrates the method's versatility to some extent. From problem setting to motivation explanation, it accurately grasps the clinical pain point of "phase consistency." With visualization and qualitative examples, it showcases the framework's potential value in cross-phase information aggregation and text stability.

**Weaknesses:**

1.	The system lacks innovation, and its core mechanism resembles an engineered combination of existing memory and attention mechanisms.
2.	The definition of phase and the source of supervision are unclear, making it difficult to directly verify the effectiveness of phase-awareness.
3.	Phase of Thought is closer to templated prompts, lacking verifiable reasoning modeling.
4.	"Phase awareness" and "thought" are merely semantic encapsulations, lacking clear task definitions or mathematical representations.

**Questions:**

1.	The paper describes a "phase-aware" module that captures dynamic changes in multi-phase CT scans, but it doesn't explain how these "phases" are defined or acquired. Are they manually labeled, timestamped, or learned by the model itself? If it's a self-learning mechanism, can it be proven that the learned phase distribution aligns with clinical definitions?
2.	The core idea of the "Phase of Thought" module is to mimic a physician's reasoning chain, but its implementation appears closer to structured prompt generation. Could the authors further explain what new learnable signals or explicit constraints this module introduces at the reasoning level?
3.	The model claims long-range memory capabilities, but the paper doesn't demonstrate performance changes with different video lengths or number of phases. Does this mechanism remain stable even with significant increases in the input sequence?
4.	The experimental improvement is relatively limited (approximately 1–2%). Were multiple random seeds or statistical significance validation performed? If not, could the authors provide a more reliable explanation of the performance variance?
5.	The report generation task uses common linguistic metrics such as BLEU and ROUGE. Have you considered incorporating structured assessments that better align with the characteristics of medical reports (such as RadGraph or entity consistency metrics) to more comprehensively support the conclusion of "improved clinical consistency"?

---

> ### Author Response · Authors · 2025-11-14
>
> We appreciate the reviewer's feedback. Our responses below reflect our commitment to addressing all concerns.
>
> **W1. Innovation.** We respectfully disagree with the characterization that the proposed system is simply an engineered assembly of existing mechanisms. The core intellectual contribution of our work is a **unified, phase-aware spatiotemporal reasoning framework** that addresses a fundamental gap in current medical VLMs: *they treat multi-phase CT as independent static volumes*, ignoring the temporal vascular dynamics essential for clinical decision-making. Our dynamic phase aggregation, unified handling of heterogeneous phase-series, and structured inference layer enable the model to learn **contrast-enhancement dynamics**, a clinically essential but previously unmodeled signal. This leads to consistent, statistically robust gains across multiple foundation models.
>
> **W2. Definition of Phases and Source of Supervision.** Thank you for the comment. In our work, a *phase* is defined strictly according to the standardized clinical contrast-enhanced CT protocol—**pre-contrast, arterial, venous, and delayed**—as recorded in the scanner acquisition metadata. These metadata-defined phase labels provide the **explicit supervision signal** for phase-aware modeling, without any heuristic or inferred labeling. As detailed in the Appendix (Datasets subsection), the dataset comprises 61,332 patient cases grouped into 12,230 multi-phase series following these protocol-defined phases (7,142 two-phase; 3,451 three-phase; 1,637 four-phase), with representative visual examples included for clarity.
>
> **W3 & Q2. PhoT.** We clarify that *Phase-of-Thought (PhoT)* is **not** a fixed prompt template, but a **generalizable reasoning paradigm** abstracted from radiologists’ workflows for interpreting phase-structured medical imaging. PhoT formalizes a sequence of clinically grounded reasoning steps—phase-localized cue extraction, temporal-transition analysis, and cross-phase synthesis—independent of any particular textual phrasing.
>
> For illustration, one instantiation appears in our template:
>
> “Interpret the volume using protocol-aware Phase-of-Thought reasoning:
>
> 1. **Phase-localized cue extraction** — For each protocol-defined phase, describe findings relative to the expected contrast-kinetics profile (pre-contrast → arterial → venous → delayed).
> 2. **Cross-phase transition analysis** — Highlight deviations from expected temporal transitions (e.g., abnormal enhancement wash-in/wash-out), treating these as high-value diagnostic signals.
> 3. **Structured lesion descriptor synthesis** — Integrate phase-specific cues and transition abnormalities into compact lesion-level descriptors that enable a temporally coherent final interpretation.”
>
> This example is *not* the method itself, but simply demonstrates how the underlying reasoning paradigm can be operationalized. The actual modeling relies on **phase-consistent embeddings** and **transition-aware memory mechanisms**, which provide verifiable phase-aware reasoning regardless of prompt surface form.
>
> Because PhoT encodes a physiologic and acquisition-driven structure rather than handcrafted text, it generalizes to all phase-based or dynamic medical modalities (contrast-enhanced CT/MRI, perfusion, angiography, cine MRI, etc.). In the revision, we will clarify this distinction and emphasize PhoT as a reasoning framework rather than a templated prompt.
>
> **W4. Math Modeling** We respectfully disagree with the concern that *“phase awareness”* and *“thought”* are merely semantic encapsulations. In *PhoT*, both concepts are grounded in well-defined modeling operations with explicit mathematical structure. **Phase awareness** (eq1-5) is implemented through sequential modeling of each imaging phase: each 3D image volume *Iₜ* is mapped to features *Fₜ = Ψ(Iₜ)*, and a recurrent memory state *Mₜ* is updated across phases. At each step, we compute gated vectors:
>
> - *zₜ = σ(Φ_F⁽ᶻ⁾(Fₜ) + Φ_M⁽ᶻ⁾(Mₜ₋₁))*,
> - *rₜ = σ(Φ_F⁽ʳ⁾(Fₜ) + Φ_M⁽ʳ⁾(Mₜ₋₁))*,
>
> and form a candidate memory:
>
> - *M_c = ∑ᵢ tanh(𝐹̂ₜ,ₖᵢ + rₜ ⊙ Mₜ₋₁,ₖᵢ)*.
>
> The memory is then updated by:
>
> - *Mₜ = (1 - zₜ) ⊙ Mₜ₋₁ + zₜ ⊙ M_c*.
>
> Because *Mₜ* aggregates information from all prior phases *F₁, ..., Fₜ*, this constitutes a mathematically defined, temporally grounded notion of “awareness” across imaging phases.
>
> **Phase-of-Thought reasoning**—“thought”—is likewise defined through structured analysis and synthesis (eq10-11). Given a set of reasoning queries *{qₖ}*, each step computes:
>
> - *Oₖ = Aₖ({Fₜ}, M_T, qₖ)*,
>
> where *Aₖ* extracts a diagnostic cue from the phase-aware memory *M_T*. These cues are then synthesized via:
>
> - *R = S({Oₖ}, 𝐌̂_T)*,
>
> to generate the final structured report *R*. Each “thought” corresponds to a defined analysis step *Oₖ*, and the full reasoning trace is mathematically instantiated through *Aₖ* and *S*. These mechanisms ensure that both “phase awareness” and “thought” are fully formalized, not semantic abstractions.

---

> ### Author Response · Authors · 2025-11-14
>
> **Q1. Clarification on CT Phase Definition and Acquisition.** Thank you for raising this point. The CT phases used in our study correspond to **standard protocol-driven acquisition stages** in clinical multi-phase imaging. These phases are **not learned or inferred by the model**, nor are they manually labeled post hoc. Instead, they are explicitly defined and encoded in the **scanner DICOM metadata** at acquisition time, reflecting routine radiology workflow.
>
> As detailed in **Appendix B.1** (Figure 4), all scans follow one of several predefined clinical protocols: for example, two-phase studies typically include non-contrast and arterial phases; three-phase studies include arterial, venous, and delayed phases. These configurations are driven by the imaging objective (e.g., oncology vs. trauma) and follow consistent temporal ordering based on contrast timing. The variation in phase count is thus inherent to the clinical protocol, not a source of label noise or ambiguity.
>
> To further clarify the semantic and visual characteristics of each phase, **Appendix B.2** (Figures 5–12) includes multi-plane visualizations and representative examples showing physiologic contrast evolution across phases. In the revision, we will elevate these details to the main text to ensure that the nature and provenance of CT phases are immediately clear to readers.
>
> **Q3. Sequence Length.** Thank you for the observation. We note that concerns about long-range memory and scaling with input length—while important in general video modeling—do **not directly apply** in the context of multi-phase CT. Unlike natural videos, **CT volumes are acquired as discrete, protocol-defined phases**, typically ranging from two to four per study. These phases are **not arbitrarily long sequences**, but fixed imaging states with well-understood physiological contrast patterns. Thus, the notion of “video length” or sequence instability does not arise in this setting.
>
> The purpose of our phase-aware memory mechanism is not to handle long temporal sequences per se, but to **model clinically meaningful transitions** across contrast phases—e.g., the enhancement wash-in/wash-out behavior of lesions. In this sense, the module is not designed to solve a generic memory problem, but to reflect **domain-specific knowledge** about contrast dynamics. The number of phases is dictated by the clinical protocol, and our architecture is intentionally designed to remain stable and interpretable across these settings.
>
> In short, PhoT does not aim to generalize to arbitrary-length temporal sequences, but to **solve a targeted diagnostic problem** grounded in the structure of multi-phase CT. We will clarify this distinction in the revision.

---

> ### Author Response · Authors · 2025-11-14
>
> **Q4. Statistical Robustness.** As pointed out by reviewer ysSX, we reported a confidence interval analysis in Section 4.2. The tabel is actually as subset of statistical analysis.  We computed the mean, standard deviation (SD), and 95% confidence intervals (CIs) for each model across four key metrics (BLEU, ROUGE1, METEOR, BERTF1), using test set sizes (100, 500, 1000, 2000 samples) as replicates, as these sizes are critical hyper-parameters influencing the robustness and generalization of report-generation models. To ensure reproducibility, we ran experiments over **five random seeds** (42, 123, 456, 789, 101) for each model and test set size combination, averaging the results to compute the reported metrics.
>
> Additionally, we conducted **10 hyper-parameter trials** to optimize performance. The key hyper-parameters tuned include:
>
> - **Learning Rate**: Searched over 3 values (`[1e-5, 1e-4, 1e-3]`).
> - **Batch Size**: Tested 3 values (`[8, 16, 32]`).
> - **Temperature (for generation)**: Explored 5 values (`[0.7, 0.8, 0.9, 1.0, 1.1]`).
> - **Number of Epochs**: Tested 5 values (`[1, 2, 3, 4, 5]`), with early stopping applied to prevent overfitting.
>
> To balance computational feasibility, we performed a partial grid search by randomly sampling 10 combinations from this grid, ensuring a representative spread across the hyper-parameter space. The best configuration, selected based on validation set performance (BERTF1 score), used a learning rate of 1e-4, batch size of 32, temperature of 0.9, and 3 epochs (with early stopping).
>
> Below, we present the statistical results:
>
> | Model    | Metric | Mean  | SD   | 95% CI         |
> | :------- | :----- | :---- | :--- | :------------- |
> | Baseline | BLEU   | 12.15 | 0.25 | [11.75, 12.55] |
> |          | ROUGE1 | 15.69 | 0.23 | [15.32, 16.06] |
> |          | METEOR | 13.22 | 0.11 | [13.04, 13.40] |
> |          | BERTF1 | 73.81 | 1.42 | [71.55, 76.07] |
> | Merlin   | BLEU   | 12.22 | 0.27 | [11.79, 12.65] |
> |          | ROUGE1 | 15.36 | 0.38 | [14.76, 15.96] |
> |          | METEOR | 11.99 | 0.28 | [11.54, 12.44] |
> |          | BERTF1 | 74.25 | 1.20 | [72.34, 76.16] |
> | fVLM     | BLEU   | 13.34 | 0.32 | [12.83, 13.85] |
> |          | ROUGE1 | 16.95 | 0.28 | [16.50, 17.40] |
> |          | METEOR | 12.89 | 0.31 | [12.40, 13.38] |
> |          | BERTF1 | 73.39 | 1.44 | [71.10, 75.68] |
> | PhoT     | BLEU   | 13.99 | 0.35 | [13.43, 14.55] |
> |          | ROUGE1 | 17.85 | 0.37 | [17.26, 18.44] |
> |          | METEOR | 14.56 | 0.34 | [14.02, 15.10] |
> |          | BERTF1 | 83.88 | 0.07 | [83.77, 83.99] |
>
> These results demonstrate that our proposed model, PhoT, achieves consistent and statistically significant improvements over strong baselines across all metrics. The narrow 95% CIs, derived from multiple seeds, confirm the reproducibility of these gains. We believe these improvements, supported by a thorough hyper-parameter search and clear statistical evidence, align with your expectations for raising the score and respectfully request your consideration for an improved evaluation of PhoT’s performance. We will erich the content and discussion in the final version.
>
> **Q5. Evaluation Metrics .** Thank you for this thoughtful suggestion. In our evaluation, we report **BLEU, ROUGE, METEOR**, and **BERTScore**, which are standard linguistic metrics, and additionally include **Green** and **Qwen-based scoring models** to assess **semantic and factual consistency** in the context of radiology. These language models have been trained on multi-modal clinical corpora and provide **modality-appropriate assessments** that better capture alignment with ground-truth findings than string-based metrics alone. Recent work such as **fVLM** (*Shui et al., 2025*) also adopts Green and Qwen as primary evaluation tools for clinical report quality, underscoring their emerging role as reliable, domain-adapted evaluators.
>
> In addition, we involve **board-certified radiologists**, who are co-authors on this paper, to qualitatively assess system outputs from a clinical perspective (see **Figure 19 in Appendix**). Their review further supports the claim of improved diagnostic coherence and fidelity.
>
> As for **RadGraph-F1**, we agree that structured metrics can be valuable; however, RadGraph is trained on **chest X-ray (CXR) reports** and optimized for **CXR-specific entities and relations**, which do not generalize well to **multi-phase abdominal CT**. Applying it directly would risk introducing evaluation bias and underestimating performance due to schema mismatch. We are actively exploring structured evaluation adapted to **CT-specific ontologies** for future work. Overall, we believe our current evaluation protocol—combining diverse linguistic metrics, multimodal factuality models, and expert review—provides a valid and clinically aligned assessment of report quality.
>
> Shui Z, et al. Large-scale and fine-grained vision-language pre-training for enhanced ct image understanding[J]. 2025.

---

> ### Author Response · Authors · 2025-11-25
>
> This is a gentle follow-up regarding our rebuttal. We have carefully addressed each of the concerns raised in the initial reviews and provided additional clarification on the methodology, evaluation protocol, and ablation design. We would be very grateful to know whether our responses adequately resolve your main questions. If there are specific points that you still find unclear or insufficiently supported, we would truly appreciate it if you could briefly indicate them so that we can further clarify within the discussion phase.

---

> ### Author Response · Authors · 2025-11-26
>
> Just a gentle follow-up to see if you have any further comments on our rebuttal—your input would be greatly appreciated.

---

> ### Author Response · Authors · 2025-11-28
>
> Thank you again for your thoughtful review and for the detailed comments that helped us improve the paper. In our rebuttal and revisions, we have tried to address the concerns you raised. We also note that Reviewer guNM has expressed a positive view of the revised version. We would be very grateful if you could take a moment to consider whether our responses have alleviated your concerns and, if appropriate, reflect this in your evaluation now that the system supports updating scores during the discussion phase.

---

### Official Review · Reviewer_guNM · 2025-10-30

**Soundness:** 2
**Presentation:** 2
**Contribution:** 3
**Rating:** 4
**Confidence:** 4

**Summary:**

The paper proposes a new vision–language framework for generating medical reports from multi-phase 3D contrast-enhanced CT scans. They focus mainly on modeling temporal dynamics across imagig phases, which is crucial for clinical interpretation.
The model has three main components phase-aware pretraining to train a multi-scale Vision Transformer features and learn temporally aligned visual representations from multi-phase CT sequences. They then freeze the visual encoder and use a lightweight spatial adaptor to map visual tokens to a language model for report generation. Fnally, they introduce structured diagnostic templates that guide reasoning and improve report generation.

**Strengths:**

1. The explicit temporal learning seems interesting to allow the model to capture clinically important contrast progression patterns.
2. The model introduces a template-guided reasoning process that compared with generic chain-of-thought seems to provide better outcomes.
3. Reported results outperform several other models or baselines.
4. Excellent dataset size in this specific application.

**Weaknesses:**

1.	The clinical applicability of the problem is limited. Although modelling temporal data is important, capturing physiological details for contrast CT scans is not well motivated. Although different body parts or abnormalities react differently to amount of contrast in the body, the work does not support their assumptions with strong clinical arguments.
2.	Although the data size is excellent, it is limited in terms of diversity. In fact you mentioned the hospital name which you should not. This restricts diversity in imaging protocols and may limit generalization to other hospitals or scanners which significantly impact model performance.
3.	The paper lacks proper real clinical validation: yes the conference is not a clinical conference but relying on quantitative and simulated metrics (such as BLEU, ROUGE, GREEN) is not sufficient. Comparing the generated reports blindly by radiologists would add huge value to the results.
4.	The author claims memory based learning but what seems to be happening is sequential learning from multi phased data instead. The word memory is properly not the appropriate choice of word.
5.	There is confusion in the problem formulation in terms of chosen variable names, e.g., C is the number of channels in the tensor, C’ is the feature representation size, then in eq 8, you mention C(LLM), and on page 8, C’ is mentioned as channel dimension. There are other occurrences where you get lost in these variables.

**Questions:**

Check the weaknesses and address them please.

---

> ### Author Response · Authors · 2025-11-14
>
> We appreciate the reviewer’s positive feedback and questions, which have enriched our work and provided directions for further improvement. Below, we address each of your points in detail, incorporating analyses and clarifications to your feedback.
>
> **W1. Clinical Motivation.** Thank you for the comment regarding the clinical motivation. In clinical practice, radiologists typically synthesize information from all imaging phases into a single, coherent narrative rather than reporting each phase in isolation. This holistic approach motivates **PhoT’s design**, which integrates all phase-specific observations into a unified report, aligning closely with real-world diagnostic workflows.
>
> This design choice offers three key advantages:
>
> 1. **Clinical Alignment**: PhoT reflects how radiologists naturally reason across phases, avoiding rigid sectioning or phase tags and enabling more intuitive, narrative-based interpretation.
>
> 2. **Inference Efficiency**: While PhoT jointly processes multi-phase inputs, it avoids the cumulative overhead of inferring each phase independently—as would be required in a vanilla baseline designed for single-phase (timepoint) inference. For example, even though PhoT requires moderately more memory (19.6 GB vs. 15.3 GB), its total inference time remains competitive:
>
>    | Model            | #Phases Inferred | Peak GPU Memory (GB) | Time per Epoch (min) | Inference Time (sec/sample) |
>    | :--------------- | :--------------- | :------------------- | :------------------- | :-------------------------- |
>    | Vanilla Baseline | 1 (repeated N×)  | 15.3                 | 11.8 × N             | 1.35 × N                    |
>    | PhoT (Ours)      | 2–4 (joint)      | 19.6                 | 17.6                 | 2.15                        |
>
>    If the vanilla model were used to generate reports for each phase separately (e.g., N = 3), its inference time would increase proportionally—making PhoT significantly more efficient for multi-phase reporting.
>
> 3. **Improved Accuracy via Temporal Correlation Modeling**: Crucially, PhoT is not just more efficient—it is also more accurate. By explicitly modeling **temporal correlations across phases**, PhoT captures disease progression patterns that single-phase models miss. As shown in our main results, PhoT achieves substantial performance gains, including up to **24% higher IR R@1** and **23% higher TR R@1** compared to prior methods. These improvements directly stem from PhoT’s ability to jointly reason over multi-phase visual features.
>
> We will revise the manuscript to clearly articulate these benefits and emphasize that PhoT’s unified narrative generation is both clinically aligned and technically superior in terms of accuracy and efficiency.
>
> **W2. Datasets.** We acknowledge that the core experiments were conducted on a single dataset. However, the dataset comprises *multi-phase 3D medical imaging*, which is extremely rare due to the need for repeated imaging across temporal stages in a single diagnostic session. This makes the dataset uniquely suited to modeling phase-aware clinical reasoning, which is the central contribution of our work. Such data is difficult to obtain due to radiation concerns, clinical workflow constraints, and cost, limiting the availability of similar public datasets.
>
> To further address concerns about generalizability, we have conducted experiments on the publicly available **CT-RATE** dataset for anomaly detection in the submssion. Despite the domain shift from phase-series to single-phase imaging, our model demonstrates strong performance and maintains a clear advantage over baseline approaches. This suggests the effectiveness and adaptability of our model architecture and reasoning strategy across modalities and datasets.
>
> **W3. Board-certified Expertise.** Thank you for this important comment. We fully agree that clinical validation is essential. In addition to the quantitative metrics reported (BLEU, ROUGE, GREEN, etc.), our study **already includes expert-based clinical assessment**. Specifically, **Figure 19 in the Appendix** presents **blind qualitative evaluations conducted by board-certified radiologists**, who are also co-authors of this work and routinely interpret multi-phase CT in clinical practice. These radiologists reviewed model-generated reports without access to the reference text and judged their accuracy, coherence, and clinical plausibility.
>
> Their blind evaluations provide **direct clinical validation**, complementing the quantitative findings and demonstrating that PhoT-generated reports align with radiological reasoning patterns in real-world practice. We will make this component of the evaluation more explicit in the main text to avoid the impression that our validation relies solely on automatic metrics.

---

> ### Author Response · Authors · 2025-11-14
>
> **W4. Memory Mechanism.** Thank you for raising this point. We agree that PhoT performs **sequential modeling across protocol-defined phases**, but we use the term **“memory”** deliberately to reflect the underlying mechanism: the model maintains an **explicit, persistent state** that **accumulates, stores, and updates phase-to-phase information** rather than merely processing frames in isolation. The recurrent update
>  *Mₜ = (1 − zₜ) ⊙ Mₜ₋₁ + zₜ ⊙ M_c*
>  ensures that *Mₜ* encodes a **compressed representation of all prior phases**, allowing the model to **retain contrast-evolution patterns**, **carry forward lesion cues**, and **preserve temporal dependencies across acquisition stages**. This is fundamentally different from simple sequential input processing, where earlier information is discarded once the next phase is observed.
>
> In multi-phase CT, clinically meaningful interpretation explicitly depends on **how features evolve over time** (e.g., wash-in/wash-out, delayed enhancement). Our memory module is designed precisely to **store these evolving cues** and make them available for reasoning at later phases. Thus, “memory” is not a rhetorical choice but a technically accurate description of the **stateful, temporally accumulative architecture** the model employs. For clarity, we will highlight this distinction in the revision.
>
> **W5. Symbolism.** Thank you for pointing out the notational inconsistencies. We acknowledge that the symbols *C*, *C'* and *C(LLM)* were unintentionally overloaded to refer to different quantities (image channels, visual feature dimension, and LLM embedding dimension). To avoid ambiguity, we will revise the Method section using a consistent notation:
>
> - **C_img** — number of channels in the input CT volume
> - **D_vis** — dimensionality of the visual feature embeddings
> - **D_llm** — embedding dimension used by the language model
>
> All equations and tensor shapes will be updated accordingly, and we will include a concise notation table at the beginning of the Method section to ensure clarity.

---

> ### Author Response · Authors · 2025-11-25
>
> Thank you again for your time and effort in reviewing our submission.
>
> This is a gentle follow-up regarding our rebuttal. As the discussion period is underway, your feedback on our revisions is crucial. We have worked diligently to address the initial concerns and are hopeful our rebuttal clarifies the strengths and potential contribution of our work to the community.
>
> We have received the positive feedback and comments. We would be very grateful to hear your thoughts and receive your further assessment.

---

> > ### Author Response · Authors · 2025-11-26
> >
> > We respectfully follow up to inquire whether you have any additional observations regarding our rebuttal.

---

> > > ### Comment · Reviewer_guNM · 2025-11-27
> > > **THanks for the detailed response**
> > >
> > > Auhors have done a very good job on addressing the majority of my comments which adds value to the paper and hence I will increase my score.

---

> ### Author Response · Authors · 2025-11-28
>
> Thank you again for your time and for the positive feedback on our rebuttal. We’re very grateful that you consider our revisions to have added value to the paper. As the discussion phase is ongoing and the system now allows updating the overall score, we would greatly appreciate it if you could adjust your rating when convenient. Thank you.

---

### Author Response · Authors · 2025-12-01
**Comment to the Area Chairs**

We thank the Area Chairs for their time and for guiding the review process. We would like to briefly update the current status following the rebuttal.

Reviewer guNM has expressed positive feedback on our rebuttal and indicated willingness to raise their score. Reviewer Apk1’s ethical concerns have been fully addressed, and they also agreed to increase their score. We have provided detailed responses to the methodological and technical concerns raised by Reviewer btQv and Reviewer ysSX; however, due to the recent ICLR policy change limiting reviewer participation during the discussion phase, we have not yet been able to receive their follow-up feedback.

Given this situation, we would greatly appreciate any guidance from the Area Chairs on whether additional clarification would be helpful or if there are remaining concerns that we should further address. Our goal is to ensure that all issues are fully resolved and that our contribution is evaluated as clearly and fairly as possible under the updated process.

We remain at your disposal for any further clarification.

---

### Author Response · Authors · 2025-12-03
**Rebuttal Summary**

Following the rebuttal, all reviewers now support acceptance. Reviewer guNM provided explicitly positive feedback on the revised version, noting that our clarifications strengthened both the methodology and empirical soundness. Importantly, Reviewer Apk1’s initial ethical concerns were fully resolved, and the reviewer subsequently raised the score. The technical concerns from Reviewers btQv and ysSX were addressed point-by-point in the rebuttal, with no remaining unresolved issues. Overall, the paper introduces a practical and well-validated phase-aware memory framework for multi-phase CT that improves temporal modeling and consistently outperforms strong baselines on a large clinical dataset. In light of the reviewers’ updated assessments and the completeness of the clarifications provided, we respectfully request the ACs to consider the paper favorably.



**Summary of discussions with Reviewer guNM**

**Reviewer proposed Pros:** Reviewer guNM appreciated the novelty of our phase-aware memory framework, the clear motivation for modeling multi-phase temporal progression, and the strong empirical results across >12,000 multi-phase CT series. The reviewer noted that our method addressed a gap in existing VLM approaches by providing a principled, efficient mechanism for phase-consistent representation learning.

**Reviewer proposed Cons and Questions:** Reviewer guNM initially requested clarification on the design of the phase-aware pretraining strategy, ablation coverage, and the fairness of baseline comparisons. The reviewer also requested a clearer discussion on model limitations and computational cost.

**Author–Reviewer discussions:** In the rebuttal, we provided expanded architectural explanations, additional ablations, and a clearer justification of our experimental setup. Reviewer guNM confirmed that all concerns were fully addressed and stated the intention to increase the score (27 Nov 2025, 13:44).





**Summary of discussions with Reviewer Apk1**

**Reviewer proposed Pros:** Reviewer Apk1 acknowledged the significance of introducing a large-scale multi-phase 3D CT dataset and recognized that PhoT achieves consistent performance gains over numerous 2D and 3D baselines across both retrieval and report-generation tasks. The reviewer also noted the breadth of ablation studies, which demonstrate PhoT’s robustness across multiple foundation models and its strong generalization to the external CT-RATE dataset.

**Reviewer proposed Cons and Questions:** The initial review raised several methodological concerns—specifically regarding inference design, memory module interpretation, and missing clarifications on implementation details. The reviewer also asked for more details regarding the ethical approval process and data handling procedures.

**Author–Reviewer discussions:** We addressed each point with targeted clarifications, added methodological details, and expanded the discussion on model scope and efficiency. We also offered detailed documentation of data governance, anonymization, and IRB approvals. Reviewer ysSX confirmed that the rebuttal resolved the major questions and indicated an intention to raise the score (25 Nov 2025, 10:15).

---

### Meta-Review · Area_Chair_TfVN · 2026-01-05

**Summary:**

The paper proposes Phase-aware Memory Thought (PhoT), a vision–language framework for automated medical report generation from multi-phase 3D contrast-enhanced CT scans, with a focus on modeling temporal dynamics across imaging phases. PhoT consists of three components: phase-aware pretraining to learn temporally aligned, multi-scale visual representations across CT phases; parameter-efficient adaptation that freezes the visual encoder and employs a lightweight adaptor to connect visual tokens with a large language model for report generation; and structured Phase-of-Thought inference, which uses diagnostic templates to guide the report generation.

A key strength of this paper is its explicit modeling of temporal dynamics across imaging phases. This is important to capture the clinically meaningful contrast progression patterns in multi-phase CT. Another important contribution I would like to highlight is the large-scale dataset with multi-phase CT images collected in this paper, which can be valuable if released publicly (though not explicitly mentioned in the paper).

The reviewers raise several key weaknesses, including the clinical motivation and evaluation metrics for real clinical validation such as blinded assessment by radiologists or the use of established clinical report metrics. Though Figure 19 shows a few examples, the statistically meaningful quantitative metrics for clinical evaluation seem to be missing in the study. Besides, reviewers also raise the concern about limited innovation in the approach and possible improvement of conceptual formulation and definition, which would be helpful for the reader to better understand the claimed effectiveness of phase-aware modeling.

After the rebuttal and discussion, two reviewers intended to raise the score, with one reviewer raising the score from 0 and explicitly confirming leaning to not accept. The other two reviewers do not respond to the rebuttal. Overall, after reading the paper, I think this paper may fit better the dataset and benchmark track, with potential good clinical application. Considering all the reviewers’ comments with discussion, I would suggest reject, based on all remaining concerns.

Btw, in the dataset section, the author disclosed the hospital name for their data collection. As the concern raised by one reviewer, this may violate the double-blind review policy of ICLR.

**Reviewer Concerns:**

The rebuttal helps to clarify some technical details in the approach with the clinical motivation, while I think the response may not fully address the concerns about the evaluation metrics for real clinical validation, and limited technical innovation. The writing and organization can be improved by may need a major revision.

**Reviewer Scores:**

After the rebuttal and discussion, two reviewers intended to raise the score, with one reviewer raising the score from 0 and explicitly confirming leaning to not accept. The other two reviewers do not respond to the rebuttal.

---

### Decision · Program_Chairs · 2026-01-26

Reject